# SANEval: Open-Vocabulary Compositional Benchmarks with Failure-mode Diagnosis

## Abstract

The rapid progress of text-to-image (T2I) models has unlocked unprecedented creative potential, yet their ability to faithfully render complex prompts involving multiple objects, attributes, and spatial relationships remains a significant bottleneck. Progress is hampered by a lack of adequate evaluation methods; current benchmarks are often restricted to closed-set vocabularies, lack fine-grained diagnostic capabilities, and fail to provide the interpretable feedback necessary to diagnose and remedy specific compositional failures. We solve these challenges by introducing **SANEval** (Spatial, Attribute, and Numeracy Evaluation), a comprehensive benchmark that establishes a scalable new pipeline for open-vocabulary compositional evaluation. SANEval combines a large language model (LLM) for deep prompt understanding with an LLM-enhanced, open-vocabulary object detector to robustly evaluate compositional adherence unconstrained by a fixed vocabulary. Through extensive experiments on six state-of-the-art T2I models, we demonstrate that SANEval's automated evaluations provide a more faithful proxy for human assessment; our metric achieves a Spearman's rank correlation with statistically different results than that of existing benchmarks across tasks of attribute binding, spatial relations, and numeracy. To facilitate future research in compositional T2I generation and evaluation, we will release the SANEval dataset and our open-source evaluation pipeline.

## 1 Introduction

The growing popularity of visual generative media has been fueled by rapid progress in text-to-image (T2I) model families such as Stable Diffusion Rombach et al. (2022), Imagen Saharia et al. (2022), GPT OpenAI et al. (2024), and Gemini Image (aka Nano Banana) Google (2025). These systems generated high adoption interest across industries ranging from marketing to entertainment. Yet they often struggle with some seemingly simple tasks, such as attribute binding on multiple objects, capturing spatial relationships, or handling numbers. One clear research bottleneck is the lack of reliable evaluation methods, as existing benchmarks remain limited in scope and fail to capture many of these challenges.

Prior research has focused on alignment scores from models such as CLIP Radford et al. (2021), DINO Caron et al. (2021), and BERT Devlin et al. (2019); however, these methods were primarily designed to capture general semantic alignment or representation quality in vision–language or language-only tasks, rather than the fine-grained compositional and perceptual requirements specific to T2I evaluation. A few works have proposed improved metrics tailored to T2I tasks Dinh et al. (2022); Ross et al. (2024), offering some robustness by being less susceptible to shortcuts such as text-only cues or copy–paste image manipulations. More recently, vision–language models (VLMs) have been employed as judges in works such as Davidsonian Scene Graphs Cho et al. (2024) and ConceptMix Wu et al. (2024a), typically through binary yes/no evaluations. While this approach improves flexibility over simple metrics like CLIPScore, it typically yields a single, uninterpretable score, failing to diagnose the specific modes of compositional failure. In contrast, benchmark-oriented efforts Ghosh et al. (2023); Huang et al. (2023; 2025) have explored non-VLM strategies, such as object detectors, to test alignment and verification in T2I models. These methods, however, are fundamentally constrained by the fixed vocabularies of their underlying object detectors (e.g., the 80 classes in MS-COCO). Ross et al. (2024). This creates a critical 'vocabulary mismatch' problem, where they cannot evaluate prompts containing any of the long tail of real-world objects.

Although no current approach offers a provable solution, recent advances in computer vision open up the opportunity to design more comprehensive benchmarks—ones that move beyond fixed vocabularies and subjective scoring toward scalable, reliable, and interpretable evaluation. Notably, prior work has largely overlooked the importance of feedback-driven insights, which are crucial for diagnosing specific failure modes. While human preferences have long been central to image generation Fu et al. (2024); Wu et al. (2024b); Zhou & Lee (2024), feedback-oriented objective evaluation at scale has yet to emerge as a core theme in benchmarking.

This work presents **SANEval**, an objective benchmark for quantifying the compositional faithfulness of T2I models. Our contributions are two-fold: (i) an open-source reasoning dataset of prompts, generated images, and diagnostic conformity labels, and (ii) a plug-and-play evaluation pipeline that is scalable, flexible, and equipped with open-world detection capabilities. At its core, SANEval consists of a *Prompt Understanding Module*, which extracts objects, attributes, and relations from text, and an *Enhanced Object Detection Module*, which combines LLMs with open-world detectors Wang et al. (2025) to robustly identify and map objects. Together, these components enable systematic evaluation across attribute binding, spatial relations, and numeracy, while generating interpretable feedback to diagnose specific failure modes.

Our evaluation framework is built around three modules—Attribute Binding, Spatial Relationships, and Numeracy—each targeting a different aspect of compositional adherence in T2I models. The attribute binding module evaluates whether object-level attributes such as color, shape, and texture are correctly rendered. The spatial relationship module focuses on positional relations between objects (e.g., *on top of*, *next to*, *between*), while the numeracy module measures whether the correct number of instances are generated. Importantly, all three modules not only assign quantitative scores but also produce conformity labels that provide objective, interpretable feedback by pinpointing what is missing, incorrect, or spurious.

Our contributions are as follows:

- We introduce **SANEval** (Spatial, Attribute, and Numeracy Evaluation), a suite of compositional benchmarks for T2I models.
- We release a diverse dataset of prompts, corresponding images from multiple state-of-the-art T2I generators, and feedback-based conformity labels.
- We propose three modules that decouple evaluation into distinct compositional axes (spatial, attribute, and numeracy) and provide both numerical scores and structured, diagnostic feedback.
- We validate our benchmark against human annotations, demonstrating strong alignment and justifying the design of our evaluation modules.

Section 2 reviews related work and preliminaries. Section 3 introduces our methodology and evaluation pipeline. Section 4 presents quantitative and qualitative results, followed by analysis. Finally, Section 5 concludes with key findings and future directions.

## 2 BACKGROUND

Despite rapid advances in T2I models, evaluation has lagged behind. Existing efforts span automated metrics, compositional adherence benchmarks, and verification paradigms such as VQA and object detection, but each suffers from limitations in scalability, interpretability, or open-world generalization. We review these directions and their shortcomings below.

### 2.1 AUTOMATED EVALUATION BENCHMARKS

Recent advances in T2I models Rombach et al. (2022); Chen et al. (2024b); Betker et al. (2023); Pramanik et al. (2025); Sun et al. (2024) have enabled the generation of high-fidelity, photorealistic images from natural language prompts. In stark contrast, evaluation pipelines have not kept pace with this rapid progress. A key challenge is the inherent subjectivity of generative AI, Zhang & Gosline (2023), which complicates the definition of objective, universal evaluation standards. Unlike traditional supervised learning tasks, where metrics such as accuracy or mean squared error are well-defined and scalable, commonly used generative metrics such as Fréchet Inception Distance (FID),

Inception Score (IS), and Kernel Inception Distance (KID) Kynkäänniemi et al. (2023) suffer from inflexibility and poor scalability. These metrics often require expensive infrastructure (e.g., GPUs) and are sensitive to factors such as sample size, making them less reliable for large-scale or fine-grained evaluation of T2I models.

Human evaluation remains the gold standard for assessing image quality; however, it is time-consuming, expensive, and difficult to standardize across studies Bakr et al. (2023). Early automated approaches measured image realism with FID Heusel et al. (2017) and text–image alignment with CLIPScore Hessel et al. (2021), but these metrics cannot capture the fine-grained compositional errors that define today's T2I challenges. Such errors include generating the wrong number of objects, misplacing spatial relations, or failing to render correct attributes. As T2I models have advanced, this gap has highlighted the need for benchmarks that directly evaluate compositional adherence to prompts.

### 2.2 Compositional Adherence Benchmarks

Early training and evaluations of T2I models relied on datasets such as CUB Birds Wah et al. (2011), Oxford Flowers Xia et al. (2017), and COCO-Captions Chen et al. (2015). While useful at the time, these datasets offered limited diversity and were prone to overfitting at scale. As the field progressed, researchers began introducing more targeted benchmarks to assess compositional aspects of T2I generation. For example, DALL-Eval Cho et al. (2023) and HE-T2I Petsiuk et al. (2022) proposed 7,330 and 900 prompts respectively, focusing on attributes such as object counts and face identification. Although these works laid the foundation for modern compositional adherence benchmarks, they rely heavily on manual curation, making them costly, prone to human inconsistencies, and ultimately difficult to scale. More recent benchmarks Huang et al. (2023; 2025); Ghosh et al. (2023); Hu et al. (2024); Li et al. (2025) evaluate a wider range of compositional attributes, including spatial relationships, colors, and shapes. Building on this direction, we focus on three broad categories of compositional adherence: (i) spatial relationships, (ii) numeracy, and (iii) attribute binding.

**Spatial relationships** capture a model's ability to render objects in correct relative positions as described in the prompt (e.g., generating "a dog to the left of a cat"). While this dimension has been studied in prior works Cho et al. (2023); Bakr et al. (2023); Huang et al. (2023), many existing approaches are brittle, are brittle, relying on fragile named entity recognition models Srinivasa-Desikan (2018) combined with a fixed dictionary of relationship words (e.g., *to the left of*, *on top of*). Such rule-based matching struggles to generalize to the wide variety of natural language expressions found in real prompts, limiting scalability.

**Numeracy adherence** is relatively straightforward and has been a core focus of many prior works Huang et al. (2025); Cho et al. (2023). It measures a model's ability to generate the exact number of object instances requested (e.g., a prompt specifying *five sheep* or *a dozen mangoes*). While this task faces similar rule-based matching issues as spatial adherence, a more significant challenge arises with less common objects (e.g., *capybara* or *albatross*), where traditional object detection pipelines often fail to reliably identify instances.

**Attribute binding** refers to the ability to correctly associate specified attributes (e.g., color, texture, shape) with their intended objects, particularly in prompts with multiple object–attribute pairs. Although this has been a principal component of several benchmarks Bakr et al. (2023); Huang et al. (2025); Cho et al. (2023); Petsiuk et al. (2022), it suffers from similar limitations as numeracy and spatial adherence. A particular difficulty lies in the high rate of false positives caused by misleading or ambiguous prompts (e.g., "a red cube and a blue sphere"), where current evaluation pipelines often conflate attribute–object associations.

### 2.3 Limitations of Existing Benchmarks

While human preferences have been central to the development of large-scale T2I models Labs et al. (2025); Deng et al. (2025); Rodriguez et al. (2025); Guo et al. (2025), most existing benchmarks rely on one of two paradigms: Visual Question Answering (VQA) or Object Detection (OD). VQA-based approaches Hu et al. (2024); Li et al. (2025); Hu et al. (2023), often built on BLIP-VQA models Li et al. (2022), evaluate adherence by querying an image with conditions and checking for correctness.

However, this paradigm offers little interpretability and limited reasoning ability, making it sensitive to question phrasing and unable to capture subtle changes Kim et al. (2025).

In contrast, OD-based methods provide bounding boxes with semantic labels, offering greater interpretability in evaluation Zang et al. (2025). A key shortcoming, however, is their limited class coverage: for instance, detectors trained on MS-COCO can only recognize 80 classes, while those trained on PASCAL-VOC are restricted to just 20 Pramanik et al. (2024). Benchmarks such as Geneval Ghosh et al. (2023) and Compbench++ Huang et al. (2025) rely on detectors like Mask2Former Cheng et al. (2022) and UniDet Liang et al. (2025), but these models often generalize poorly and lag behind the performance of state-of-the-art YOLO-based detectors Wang et al. (2025); Tian et al. (2025). YOLO-based detectors further benefit from strong community support and active development, making them particularly well-suited for plug-and-play benchmarking frameworks. This enables greater flexibility, scalability, and robustness—qualities that are essential for any modern benchmark methodology.

Both VQA- and OD-based verification methodologies have their own advantages and disadvantages, which are largely complementary in nature: VQA approaches—and more recently their VLM-based extensions—offer flexibility in handling natural language queries but suffer from low interpretability and sensitivity to phrasing. In contrast, OD-based methods provide structured outputs with bounding boxes and labels that enhance interpretability, but remain limited by fixed vocabularies and poor open-world generalization. This work focuses on efficient hybrid approaches that combine the reasoning strengths of VLMs with the structured outputs of ODs to enable benchmarks that are reliable, scalable, and flexible for large-scale T2I models. Notably, there is also a lack of benchmarks that provide objective feedback—identifying what is missing, what is extra, or which objects and attributes fail to match the prompt—an essential component for diagnosing model behavior and potentially guiding improvement

## 3 THE SANEVAL FRAMEWORK

We first detail the dataset curation process before describing the evaluation methodology. The framework is composed of two core technical modules: a robust prompt understanding module (Section 3.2) and an enhanced object detection module (Section 3.3). These foundational modules are then leveraged to implement three distinct benchmarks that assess attribute binding, spatial reasoning, and numeracy, which are detailed in Sections 3.4 through 3.6, respectively. A top-level view of our benchmarking pipeline can be seen in Fig. 1.

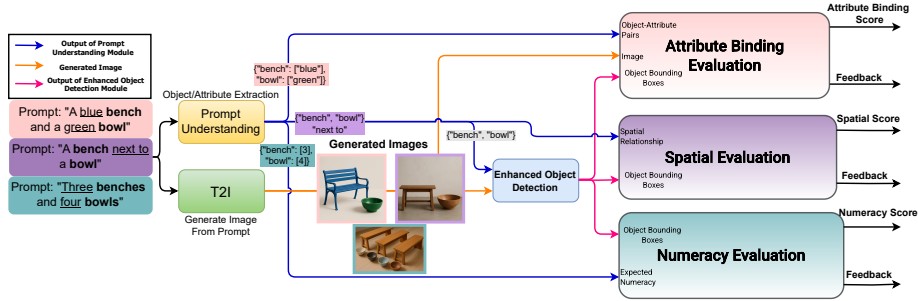

Figure 1: Block diagram description of the SANEval benchmark

### 3.1 DATASET CURATION

We open-source a sufficiently large dataset designed to evaluate compositional adherence in T2I models. The dataset consists of ~5,000 prompts, images generated from multiple state-of-the-art T2I models (5,000 generations per model), and corresponding structured feedback annotations. To ensure quality, prompts and feedback were generated using Gemini-2.5-Flash and subsequently validated by human annotators. The dataset is organized into three categories of compositional adherence:

**Attribute Binding**   This category focuses on correctly associating attributes such as color, shape, and texture with specific objects. We curate 3,000 prompts (1,000 per attribute type), each annotated with expected object–attribute bindings and feedback for cases such as attribute swaps, omissions, or hallucinations. Prompts were constructed using a script that systematically combined attribute–object pairs to ensure broad coverage.

**Spatial Relationships**   This category tests whether models can generate objects in correct relative positions (e.g., *"a dog to the left of a cat"*). We curate 1,000 prompts spanning binary and multi-object relationships, annotated with expected spatial relations and feedback for missing, incorrect, or spurious placements.

**Numeracy**   This category evaluates whether models can generate the correct number of instances specified in the prompt (e.g., *"five apples on a table"*). We curate 1,000 prompts covering small numbers as well as larger quantities, including rare object categories. Feedback annotations indicate under-generation, over-generation, and misidentification errors.

**Prompt Cardinality Breakdown**   Across all three categories, prompts are stratified by the number of objects specified: 275 prompts with 1 object, 275 with 2 objects, 275 with 3–5 objects, 125 with 6–10 objects, and 50 with 11–15 objects.

### 3.2   PROMPT UNDERSTANDING MODULE

Given a natural language prompt, the first step is to derive a structured representation of the specified content. We employ an LLM guided by carefully designed system prompts (Appendix C) to parse each input. The model extracts all objects, their attributes, spatial relationships, and numeracy. This structured representation forms the foundation of our scoring framework.

The extracted output is formatted as key–value pairs, where each key denotes a unique object and its value is a list of attributes (e.g., color, shape, texture). Spatial relations are represented as triplets of the form `<object_1, relation, object_2>` (e.g., `<dog, left_of, cat>`), allowing us to capture a wide range of relations such as *top of*, *inside*, or *next to*. Numeracy is represented directly, with each object mapped to its expected count. This module is designed to be robust to grammatical errors, stylistic variations, and complex sentences with multiple objects and attributes.

### 3.3   ENHANCED OBJECT DETECTION MODULE

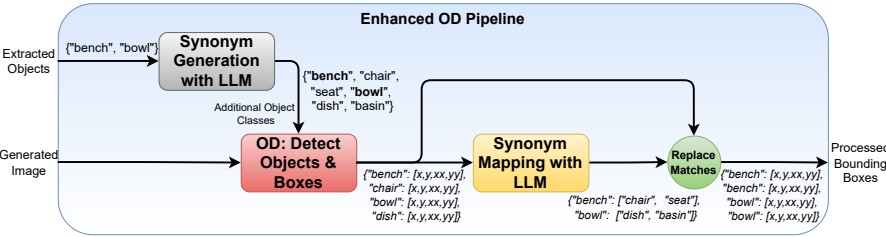

Figure 2: Block diagram of the enhanced object detection (OD) pipeline for the SANEval benchmark.

As discussed in Section 2.3, prior benchmarks have relied on either VQA- or OD-based approaches. VQA methods (and their VLM extensions) benefit from large-scale pretraining and broad world knowledge Touvron et al. (2023), but they lack interpretability and are highly sensitive to phrasing. In contrast, OD methods provide structured outputs (bounding boxes and semantic labels) that enable direct reasoning over generated content. A key requirement for benchmarking, however, is the ability to detect objects in the wild, without being constrained to fixed vocabularies.

To this end, we adopt YOLO-E Wang et al. (2025) as our primary detector, chosen for its decoupled prompt encoder that supports arbitrary object queries beyond closed-class vocabularies. This makes

it particularly suitable for open-world evaluation, where prompts frequently involve rare or domain-specific objects. While YOLO-E is our main choice, the framework is modular and can be extended to other detectors.

We further strengthen the pipeline by integrating LLM reasoning. From the Prompt Understanding Module (Section 3.2), we obtain the list of expected objects. For each, we prompt an LLM to expand the query into multiple synonyms and related terms. In parallel, YOLO-E is applied to detect objects in the generated image. Detected labels are then aligned with the expanded synonym sets, producing a robust final output that reduces vocabulary mismatches.

For example, given the prompt *"an albatross standing on a rock"*, the expected object list is {albatross, rock}. The LLM expands *albatross* into synonyms such as {albatross, seabird, large bird}. YOLO-E detects *bird* in the image. Mapping this detection back to the synonym set allows the system to correctly verify the presence of the albatross. This hybrid approach leverages the reasoning flexibility of VLMs with the structured interpretability of OD, yielding a scalable and reliable open-world benchmarking pipeline (Fig. 2).

## 3.4 ATTRIBUTE BINDING EVALUATION

Both the Prompt Understanding Module and the Enhanced Object Detection Module are shared across all three evaluation components. The attribute binding benchmark evaluates a model's ability to associate attributes with their corresponding objects, divided into three subtypes: color, shape, and texture (Fig. 4). To compute scores, we first crop the detected objects from the Enhanced Object

Figure 3: Example of the Attribute Binding evaluation pipeline in SANEval. Objects are first detected and cropped, then evaluated by a VLM on the correctness of their attributes. The upper example fully adheres to the prompt ("pink stop sign and orange bird"), yielding a perfect score of 1.0 with no feedback. The lower example partially fails ("yellow peach and a lime ceiling"), where the ceiling is missing, resulting in diagnostic feedback and a lower score of 0.5

Detection Module. For each cropped object, we then prompt an LLM to generate a targeted question about the attribute under evaluation. For example, given a prompt specifying *"a beige dress"*, the system generates the question: *"What is the color of the dress?"*—ensuring the evaluation is not biased by leaking the expected attribute. The cropped object and the generated question are passed to a VLM-as-a-judge, consistent with recent work Chen et al. (2024a); Lee et al. (2024), which outputs the detected attribute. This approach leverages the VLM's world knowledge to provide nuanced attribute detection that is robust to variations in lighting, style, and object orientation. This output is then compared against the expected attribute using an LLM that assigns a continuous similarity score between $0$ (no match) and $1$ (perfect match). For example, if the expected attribute is *blue* and the VLM predicts *turquoise*, the system assigns a score between $0.5$ and $1$, reflecting partial similarity. If the expected attribute is *beige* but the system outputs *gray*, the score is closer to $0$, indicating poor binding.

Finally, we convert these scores into structured, interpretable feedback. Instead of vague flags, the system reports issues in a natural way—for instance: *"Poor attribute binding: expected beige dress, but detected gray dress"*, or *"Partial binding: expected blue sphere, but detected turquoise sphere."* If a score falls below $0.5$, we additionally flag the attribute as missing along with the expected attribute. This two-stage mechanism ensures both an objective score for benchmarking and actionable feedback that highlights specific attribute-level deficiencies in the generated image. The

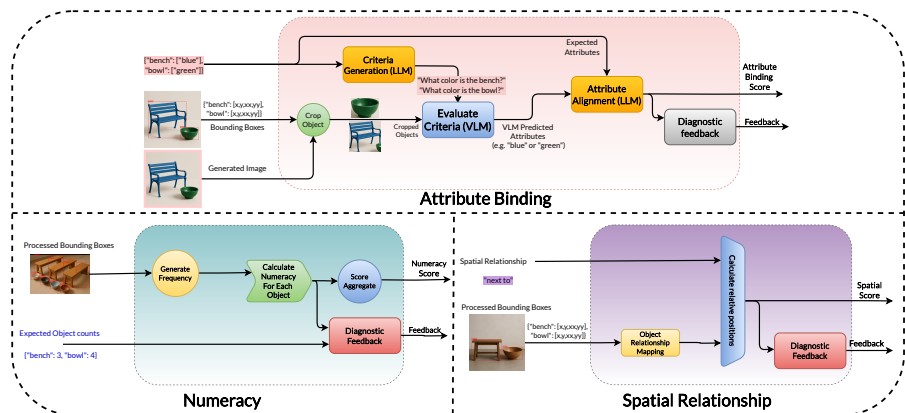

Figure 4: Block diagram descriptions of the three different scorers for the SANEval benchmark.

attribute binding benchmark quantifies a model's ability to correctly associate specified attributes with their corresponding objects, which can be seen in Fig. 4. The evaluation proceeds as follows: for each object detected in the generated image, the region defined by its bounding box is cropped. This sub-image is then passed to a Vision Language Model (VLM), in our case Gemini 2.5 Flash, which is queried to identify the relevant attribute (e.g., "What is the color of the bench?").

To score results, we employ an LLM-based evaluator that compares the predicted attribute with the ground-truth attribute, assigning a semantic similarity score on a continuous scale from $0.0$ (no match) to $1.0$ (perfect match). This avoids bias from directly revealing the expected attribute while still granting partial scores when predictions are close (*blue* vs. *cyan*). The procedure is applied to every object–attribute pair, and the final score for a prompt–image pair is the mean of all individual scores.

## 3.5 SPATIAL RELATIONSHIP EVALUATION

The spatial relationship benchmark evaluates a model's ability to render objects in correct relative positions. We follow the methodology of Huang et al. (2025), but extend it in several key ways. Specifically, we categorize eight distinct spatial relations (e.g., *to the left of*, *above*, *behind*), with the full list provided in the Appendix. Unlike Huang et al. (2025), which relies on named entity recognition and fixed lexical cues, our approach leverages the structured output of the Prompt Understanding Module, avoiding brittle dictionary-based matching and enabling robustness to diverse natural language prompts.

The evaluation uses bounding boxes from the Enhanced Object Detection Module (Section 3.3) together with spatial relations extracted by the Prompt Understanding Module (Section 3.2). For each relation, we apply a geometric evaluation function (e.g., verifying whether the centroid of *object A* lies to the left of *object B*) to compute a spatial adherence score. Feedback is generated by comparing expected versus detected objects, allowing explicit reports of errors such as *"Expected cat to the right of dog: Missing object: cat"*. This design combines objective geometric reasoning with interpretable, fine-grained feedback, yielding both reliable scoring and actionable diagnostics for spatial errors in T2I models.

## 3.6 NUMERACY EVALUATION

The numeracy benchmark evaluates a model's ability to generate the correct number of object instances specified in the prompt. Target counts are extracted by the Prompt Understanding Module and compared to detections from the Enhanced Object Detection Module (Fig. 4). A score of $1.0$ is assigned for an exact match, $0.5$ if the object appears but with an incorrect count, and $0.0$ if it is missing. The final image-level score is the average across all prompt objects.

In addition to scores, this module produces structured feedback by explicitly reporting over- and under-generation. For example, given the prompt *"three suitcases and two people"*, if five suitcases

and only one person are detected, the system outputs *"two extra suitcases detected; one person missing."* While numeracy evaluation is conceptually straightforward, it becomes challenging for rare categories (e.g., *capybara*, *albatross*), where traditional OD often fails to recognize instances. By leveraging open-world detection and synonym expansion, our framework mitigates these limitations and improves robustness.

## 4 RESULTS AND ANALYSIS

Our results proceeds in three stages: (i) evaluating six state-of-the-art models on SANEval to establish a performance baseline, (ii) probing their compositional breaking points under increasing prompt cardinality, and (iii) statistically validating and differentiating our benchmark from Huang et al. (2025) to demonstrate its usefulness.

### 4.1 QUANTITATIVE ANALYSIS ON SANEVAL BENCHMARKS

We quantitatively evaluated six state-of-the-art image generators with SANEval (Table 1). Results show no single model dominates; instead, strengths are specialized. Imagen 4.0 Ultra achieved the highest overall score, excelling in numeracy and fine-grained bindings such as shape and texture. Seedream 3.0, only slightly behind overall, outperformed in spatial reasoning and color binding. This divergence suggests that different architectures and training strategies cultivate distinct compositional skills. While Imagen 4.0 Ultra and Seedream 3.0 are the front-runners, other models also

Table 1: Combined results for SANEval benchmarks.

| Model | Spatial | Numeracy | Attribute-Binding | | | Averaged |
| | | | Color | Shape | Texture | |
| --- | --- | --- | --- | --- | --- | --- |
| Imagen 3.0 | 0.3524 | 0.5779 | 0.5780 | 0.2499 | 0.4272 | 0.4371 |
| Imagen 4.0 | 0.3362 | 0.5192 | 0.5453 | 0.2704 | 0.4339 | 0.4210 |
| Imagen 4.0 Ultra | 0.4222 | **0.6087** | 0.6090 | **0.3521** | **0.5041** | **0.4992** |
| Nano Banana | 0.4076 | **0.6087** | 0.5927 | 0.2827 | 0.4800 | 0.4743 |
| Seedream 3.0 | **0.4636** | 0.5900 | **0.6334** | 0.2805 | 0.4969 | 0.4929 |
| GPT Image 1 | 0.4372 | 0.5966 | 0.5850 | 0.2972 | 0.4655 | 0.4763 |

exhibit notable capabilities, such as Nano Banana matching the top score in numeracy. Although the results show a clear advancement in capabilities, the scores across all models, particularly for shape binding, are far from perfect. This finding underscores that complex compositional reasoning—especially the precise binding of multiple attributes in multifaceted scenes—remains a primary area for future research in T2I synthesis.

### 4.2 ANALYSIS OF THE COMPOSITIONAL LIMITS OF IMAGE GENERATORS

To assess compositional robustness, we analyze attribute binding (color, shape, texture) as a function of object cardinality using SANEval (Table 2). The scores deteriorates consistently as prompt complexity increases, revealing a core weakness in current models. Shape binding is the most brittle, color the most robust, and texture intermediate. For instance, Imagen 3.0's shape accuracy falls from 0.3159 on single-object prompts to 0.0612 with more than ten objects, underscoring geometric integrity as a primary failure point in crowded scenes. The rate of degradation under compositional complexity varies across models, separating the robust from the fragile. Seedream 3.0 and Imagen 4.0 Ultra are most stable: Seedream excels in color binding for high-cardinality prompts, while Imagen 4.0 Ultra is strongest overall, degrading gracefully in the challenging shape binding task. In contrast, models such as Imagen 3.0 and 4.0 show significant fragility, with performance collapsing as object count rises. Nano Banana and GPT Image 1 also struggle to scale, failing on complex compositions. These findings underscore that robustness to compositional complexity—rather than peak performance on easy prompts—is one key differentiator for state-of-the-art T2I systems.

### 4.3 COMPARISON WITH EXISTING BENCHMARKS

To validate SANEval against existing methods, we conducted a Spearman rank correlation analysis Spearman (1904) comparing its scores with those from CompBench++ Huang et al. (2025). The

Table 2: Results for SANEval Attribute Binding across Color, Shape, and Texture. Higher is better. Best scores per column are in bold

| Attribute | Model | Averaged | 1 Obj | 2 Obj | 3–5 Obj | 6–10 Obj | >10 Obj |
|---|---|---|---|---|---|---|---|
| Color | Imagen 3.0 | 0.5780 | 0.5913 | **0.6293** | 0.6013 | 0.5001 | 0.2548 |
| | Imagen 4.0 | 0.5453 | 0.6013 | 0.5704 | 0.5178 | 0.4972 | 0.3364 |
| | Imagen 4.0 Ultra | 0.6090 | 0.6597 | 0.6025 | 0.5965 | 0.5788 | 0.5037 |
| | Nano Banana | 0.5927 | **0.6667** | 0.6089 | 0.5911 | 0.4716 | 0.3890 |
| | Seedream 3.0 | **0.6334** | 0.6343 | 0.6163 | **0.6647** | **0.6310** | **0.5244** |
| | GPT Image 1 | 0.5850 | 0.6232 | 0.5885 | 0.5815 | 0.5427 | 0.4492 |
| Shape | Imagen 3.0 | 0.2499 | 0.3159 | 0.2571 | 0.2394 | 0.1643 | 0.0612 |
| | Imagen 4.0 | 0.2704 | 0.3647 | 0.2954 | 0.2229 | 0.1538 | 0.0957 |
| | Imagen 4.0 Ultra | **0.3521** | **0.4311** | **0.3330** | **0.3339** | **0.2877** | **0.2195** |
| | Nano Banana | 0.2827 | 0.3649 | 0.2935 | 0.2306 | 0.1805 | 0.1198 |
| | Seedream 3.0 | 0.2805 | 0.3195 | 0.2702 | 0.2810 | 0.2342 | 0.1997 |
| | GPT Image 1 | 0.2972 | 0.4106 | 0.3094 | 0.2519 | 0.1650 | 0.1048 |
| Texture | Imagen 3.0 | 0.4272 | 0.4090 | 0.4550 | 0.4108 | 0.4025 | 0.2836 |
| | Imagen 4.0 | 0.4339 | 0.4663 | 0.4595 | 0.4035 | 0.3243 | 0.2353 |
| | Imagen 4.0 Ultra | **0.5041** | 0.5471 | **0.5204** | 0.4610 | **0.4509** | **0.3589** |
| | Nano Banana | 0.4800 | **0.5731** | 0.4964 | 0.4142 | 0.3472 | 0.2767 |
| | Seedream 3.0 | 0.4969 | 0.5442 | 0.4941 | **0.4836** | 0.4199 | 0.3246 |
| | GPT Image 1 | 0.4655 | 0.4378 | 0.5158 | 0.4437 | 0.4094 | 0.3261 |

resulting p-values, summarized in Table 3, test the null hypothesis of no monotonic relationship between benchmark scores ($\alpha = 0.05$). In most categories, such as spatial and numeracy, p-values are extremely small (often $p < 10^{-20}$). These highly significant results, combined with low correlation coefficients, indicate that while the benchmarks' rankings are not fully independent, the overlap is weak. Thus, SANEval and CompBench++ capture different signals of model performance and probe distinct parts of compositional reasoning. Notably, some categories diverge strongly, with p-

Table 3: Spearman correlation p-values comparing SANEval and CompBench++ methods

| Model | Spatial | Numeracy | Attribute-Binding | | |
|---|---|---|---|---|---|
| | | | Color | Shape | Texture |
| Imagen 3.0 | $2.639 \times 10^{-55}$ | $2.442 \times 10^{-34}$ | $9.340 \times 10^{-14}$ | $1.434 \times 10^{-1}$ | $6.311 \times 10^{-3}$ |
| Imagen 4.0 | $3.273 \times 10^{-51}$ | $1.948 \times 10^{-30}$ | $1.252 \times 10^{-13}$ | $1.695 \times 10^{-3}$ | $1.895 \times 10^{-8}$ |
| Imagen 4.0 Ultra | $2.458 \times 10^{-43}$ | $1.004 \times 10^{-20}$ | $2.565 \times 10^{-13}$ | $6.596 \times 10^{-2}$ | $5.760 \times 10^{-3}$ |
| Nano Banana | $8.178 \times 10^{-40}$ | $2.971 \times 10^{-29}$ | $1.864 \times 10^{-15}$ | $7.144 \times 10^{-3}$ | $6.355 \times 10^{-8}$ |
| Seedream 3.0 | $8.819 \times 10^{-46}$ | $1.228 \times 10^{-30}$ | $8.845 \times 10^{-16}$ | $2.728 \times 10^{-2}$ | $7.773 \times 10^{-1}$ |
| GPT Image 1 | $7.223 \times 10^{-39}$ | $3.793 \times 10^{-37}$ | $5.366 \times 10^{-11}$ | $3.413 \times 10^{-4}$ | $1.447 \times 10^{-3}$ |
| **Averaged** | $1.340 \times 10^{-39}$ | $1.673 \times 10^{-21}$ | $9.023 \times 10^{-12}$ | $3.097 \times 10^{-2}$ | $1.318 \times 10^{-1}$ |

values above the significance threshold. For example, in shape binding for certain models and in the overall texture category ($p = 0.132$), we fail to reject the null hypothesis, suggesting these attributes are evaluated so differently as to be effectively independent. Overall, the analysis provides strong statistical evidence that SANEval delivers a complementary evaluation, introducing novel criteria that address key gaps in existing compositional assessment methodologies.

## 5 CONCLUSIONS

We introduced SANEval, a benchmark addressing key gaps in T2I evaluation: reliance on fixed vocabularies and lack of diagnostic feedback. By using our proposed prompt understanding module and enhanced object-detection module, SANEval enables nuanced assessment of compositional adherence. Our contributions include an open-source dataset with diagnostic labels, a framework evaluating attribute binding, spatial relations, and numeracy, and validation on six SOTA models. Beyond offering finer-grained evaluation, SANEval provides actionable signals for RL with AI feedback, transforming evaluation into supervision. It thus establishes a foundation for feedback-driven, open-world benchmarking, paving the way toward more robust and controllable generative models across modalities.

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

## APPENDIX

## A  USE OF LLMs

Large Language Models (LLMs) were employed in a supporting role across different stages of this work. During dataset construction and experimentation, LLMs assisted with coding tasks such as prompt generation, data formatting, and automation scripts, which accelerated the development pipeline but did not replace human design or verification. In addition, LLMs were occasionally used to draft alternative phrasings or refine sections of the manuscript, with the goal of improving readability, flow, and consistency of style.

It is important to note that the role of LLMs was strictly limited to auxiliary support. All scientific ideas, experimental design, methodological innovations, and analyses were conceived, implemented, and validated by the authors. Any LLM-generated content—whether in code or text—was subject to careful review, editing, and verification to preserve the accuracy, originality, and integrity of our contributions. Thus, while LLMs served as helpful tools for productivity and clarity, the intellectual and technical content of this paper remains entirely the work of the authors.

# B  INFRASTRUCTURE AND SUPPORT

Unless otherwise noted, both the Vision-Language Model (VLM) and Large Language Model (LLM) components in our framework default to Gemini-2.5-Flash. All experiments were conducted on cloud infrastructure provisioned through Amazon Web Services (AWS). Specifically, we use the Elastic Container Service (ECS) with the Fargate launch type, which allows jobs to be deployed in a serverless fashion on top of pre-configured Docker images. These images were custom-built and reused across multiple runs to ensure consistency, reproducibility, and efficient resource management.

Our compute configuration typically consists of Linux-based operating systems running on the ARM64 architecture, with containers provisioned with 8 virtual CPUs, 12 GB of RAM, and up to 48 GB of disk storage. This setup provides a balanced environment suitable for large-scale evaluation while remaining cost-effective.

For executing inference with VLMs, LLMs, and text-to-image generation models, we rely on API endpoints provided by their respective service providers. In particular, we use endpoints from OpenAI, Google, and ByteDance (via the Fal-AI platform). This API-based design allows us to access state-of-the-art models without the need for extensive local deployment and ensures that our benchmarking pipeline remains modular and easily extendable as newer models become available.

# C  PROMPTS

This section lists the exact prompts used for LLMs and VLMs in our framework. We report them verbatim to ensure clarity and reproducibility. Note that additional post-processing steps such as parsing, schema enforcement, and consistency checks were applied during evaluation, but those details are beyond the scope of this section.

---

**Prompt 1:  Used for generating objects from a prompt**

```
Extract all significant objects (nouns) from the following text
prompt.
Text Prompt:  "{prompt}
    • Identify all significant objects (nouns) mentioned in the
      prompt, including:
        – Living beings (people, animals, creatures)
        – Substantial physical objects (vehicles, furniture,
          buildings, tools, toys)
        – Natural features (trees, rocks, mountains, water bodies)
        – Important branded items or recognizable objects
    • Do not include:
        – Positional/directional terms (top, bottom, left, right,
          front, back, side, center)
        – Clothing items (shirts, pants, shoes, hats) unless they
          are the main focus
        – Body parts (hands, face, legs) unless they are the main
          focus
        – Abstract concepts (time, love, happiness)
        – Very small or insignificant items (buttons, zippers,
          laces)
        – Prepositions or spatial relationships (on, in, under,
          above, below)
    • Return the result as a JSON array containing object names in
      lowercase, singular form.
  For example:
      • "a red car and blue bike" → ["car", "bike"]
      • "large wooden table with metal legs" → ["table"]
```

---

- "small white dog running in the park" → ["dog", "park"]
- "three books on the shelf" → ["book", "shelf"]

**Prompt 2: Used for spatial relationship**

Extract the spatial relationship and objects from the following text prompt.
Text Prompt: "{prompt}
Available spatial relationships: {available_relationships}

- Identify:
    - The first object (obj1) - the reference object
    - The spatial relationship - must be exactly one from the available relationships
    - The second object (obj2) - the object being positioned relative to obj1
- Return the result as a JSON object with exactly these keys:
    - "obj1": the first/reference object (lowercase, singular form)
    - "relationship": the spatial relationship (exactly as written in available relationships)
    - "obj2": the second object (lowercase, singular form)
- If no valid spatial relationship is found, return an empty object: {}
- Focus on the main spatial relationship in the prompt

For example:

- "a red car on the left of a blue bike" → {"obj1": "car", "relationship": "on the left of", "obj2": "bike"}
- "dog next to the table" → {"obj1": "dog", "relationship": "next to", "obj2": "table"}
- "cat on top of the chair" → {"obj1": "cat", "relationship": "on top of", "obj2": "chair"}

**Prompt 3: Used for object-attribute generation**

Extract objects and their attributes from the following text prompt.
Text Prompt: "{prompt}

- Identify all objects (nouns) and their associated attributes (adjectives or descriptive words).
- Return the result as a JSON object where:
    - Keys are object names (*lowercase, singular form*)
    - Values are lists of attributes associated with each object
- Only include objects that have explicit attributes mentioned.
- If an object has no attributes, don't include it.

For example:

- "a red car and blue bike" → {"car": ["red"], "bike": ["blue"]}
- "large wooden table" → {"table": ["large", "wooden"]}
- "small white dog running" → {"dog": ["small", "white"]}

**Prompt 4: Used for generating numeracy**

Extract all objects and their exact quantities from the following
text prompt.
Text Prompt: "{prompt}
Available number words: {available_numbers}

- Identify all objects (nouns) and their associated quantities
  (numbers).
- Include both written numbers (*e.g., "seven"*) and numeric
  digits (*e.g., "7"*).
- Return a JSON object with exactly these keys:
    - "objects": a dictionary mapping object names to their
      quantities as integers
    - "numbers_found": a list of the number words/digits found
      in the prompt

For example:

- "seven horses and 4 pigs" → {"objects": {"horse": 7,
  "pig": 4}, "numbers_found": ["seven", "4"]}
- "three red cars and two blue bikes" → {"objects": {"car":
  3, "bike": 2}, "numbers_found": ["three", "two"]}
- "a single dog and five cats" → {"objects": {"dog": 1,
  "cat": 5}, "numbers_found": ["a", "five"]}

Conversion rules:

- "one"/"a"/"an"/"single" = 1
- "two"/"couple" = 2
- "three" = 3
- "four" = 4
- "five" = 5
- "six" = 6
- "seven" = 7
- "eight" = 8
- "nine" = 9
- "ten" = 10
- "dozen" = 12
- "hundred" = 100

Guidelines:

- Convert object names to *lowercase, singular form.*
- Only include objects that have explicit or implied
  quantities.
- If no specific number is mentioned but the object appears in
  a counting context, assume quantity 1.
- Focus on the main objects being counted in the prompt.

**Prompt 5: Used for generating synonyms**

Analyze the following list of objects and identify any synonyms or
similar objects that should be grouped together.
Objects: {object_names}

- For each group of synonymous objects, choose the most
  common/standard term to represent the group.
- Return the result as a JSON object where:

```
        – Keys are the standard/representative terms (lowercase,
          singular form).
        – Values are arrays of all synonymous terms that should be
          grouped under that key.
For example:
    • ["boy", "child", "person", "girl"] → {"person": ["boy",
      "child", "person", "girl"]}
    • ["house", "home", "building"] → {"house": ["house", "home",
      "building"]}
    • ["car", "vehicle", "automobile"] → {"car": ["car",
      "vehicle", "automobile"]}
    • ["dog", "cat", "bird"] → {"dog": ["dog"], "cat": ["cat"],
      "bird": ["bird"]}
    • ["bee", "bird", "animal"] → {"bee": ["bee"], "bird":
      ["bird", "animal"]}
    • ["desk", "table", "furniture"] → {"desk": ["desk",
      "table"], "furniture": ["furniture"]}
Only group objects that are truly synonymous or refer to the same
type of thing.  Keep unrelated objects separate.
```

**Prompt 6:  Used for mapping synonyms**

```
Analyze the following list of objects and identify any synonyms or
similar objects that should be grouped together.
Objects:  {object_names}
    • For each group of synonymous objects, choose the most
      common/standard term to represent the group.
    • Return the result as a JSON object where:
        – Keys are the standard/representative terms (lowercase,
          singular form).
        – Values are arrays of all synonymous terms that should be
          grouped under that key.
For example:
    • ["boy", "child", "person", "girl"] → {"person": ["boy",
      "child", "person", "girl"]}
    • ["house", "home", "building"] → {"house": ["house", "home",
      "building"]}
    • ["car", "vehicle", "automobile"] → {"car": ["car",
      "vehicle", "automobile"]}
    • ["dog", "cat", "bird"] → {"dog": ["dog"], "cat": ["cat"],
      "bird": ["bird"]}
    • ["bee", "bird", "animal"] → {"bee": ["bee"], "bird":
      ["bird", "animal"]}
    • ["desk", "table", "furniture"] → {"desk": ["desk",
      "table"], "furniture": ["furniture"]}
Only group objects that are truly synonymous or refer to the same
type of thing.  Keep unrelated objects separate.
```

**Prompt 7:  VLM Prompt for criteria evaluation**

```
    • Color:  "What color is the object?"
    • Shape:  "What shape is the object?"
```

```
        • Texture:  "What texture does the object have?"
```

```
 Prompt 8:  Used for evaluating attribute alignment

Use LLM to evaluate how well the response matches the expected
attribute.
Args:
    • response:  VLM response describing the object
    • expected_attribute:  The attribute we're looking for
Returns:  Score between 0.0 and 1.0
Prompt:
    You are evaluating whether a description matches a
    specific attribute.
    Expected attribute:  "{expected_attribute}"
    Description:  "{response}"
    Rate how well the description matches the expected
    attribute on a scale from 0.0 to 1.0:
      • 1.0:  Perfect match (the description clearly
        contains or describes the expected attribute)
      • 0.8--0.9:  Very good match (the description
        strongly suggests the expected attribute)
      • 0.5--0.7:  Partial match (the description somewhat
        relates to the expected attribute)
      • 0.2--0.4:  Weak match (there might be some
        connection but it's unclear)
      • 0.0--0.1:  No match (the description doesn't
        relate to the expected attribute at all)
    Consider semantic similarity, synonyms, and related
    concepts.  For example:
      • "red" matches "crimson" or "scarlet" (0.8--0.9)
      • "large" matches "big" or "huge" (0.8--0.9)
      • "wooden" matches "made of wood" (1.0)
      • "smooth" might partially match "polished"
        (0.6--0.7)
    Respond with only a number between 0.0 and 1.0 (e.g.,
    "0.8").
```

## D  HUMAN EVALUATION

Table 4: Human evaluation results for spatial, numeracy, color-binding, shape-binding, and texture-binding.

| Model | Spatial | Numeracy | Attribute-Binding | | | Total |
| --- | --- | --- | --- | --- | --- | --- |
| | | | Color | Shape | Texture | |
| Imagen 3.0 | 0.8160 | 0.7720 | 0.8400 | 0.5000 | 0.7240 | 0.7304 |
| Imagen 4.0 | 0.7800 | 0.6380 | 0.7990 | 0.5170 | 0.7410 | 0.6950 |
| Imagen 4.0 Ultra | **0.9390** | 0.8870 | **0.9680** | 0.8110 | **0.9190** | **0.9048** |
| Nano Banana | 0.9220 | 0.8680 | 0.9050 | 0.6810 | 0.8150 | 0.8382 |
| Seedream 3.0 | 0.9070 | 0.7430 | 0.9170 | 0.4310 | 0.6540 | 0.7304 |
| GPT Image 1 | 0.9250 | **0.8940** | **0.9680** | **0.8530** | 0.8430 | 0.8966 |

To complement this analysis, we conducted a human evaluation (Table 4), which is an aggregation of 100 prompt adherence responses for each benchmark (500 total). Human ratings showed a strong

positive correlation with the SANEval scores in terms of overall model ranking—confirming Imagen 4.0 Ultra as a top performer and validating SANEval as a reliable indicator of general capability. However, we observed divergences at the category level. For instance, humans rated GPT Image 1 highest for numeracy and shape binding, categories where Imagen 4.0 Ultra led in the automated results. These differences suggest that human perception can prioritize different aspects of compositional accuracy and that certain artistic styles may be challenging for object detectors to parse, making human evaluation essential for a comprehensive performance assessment.

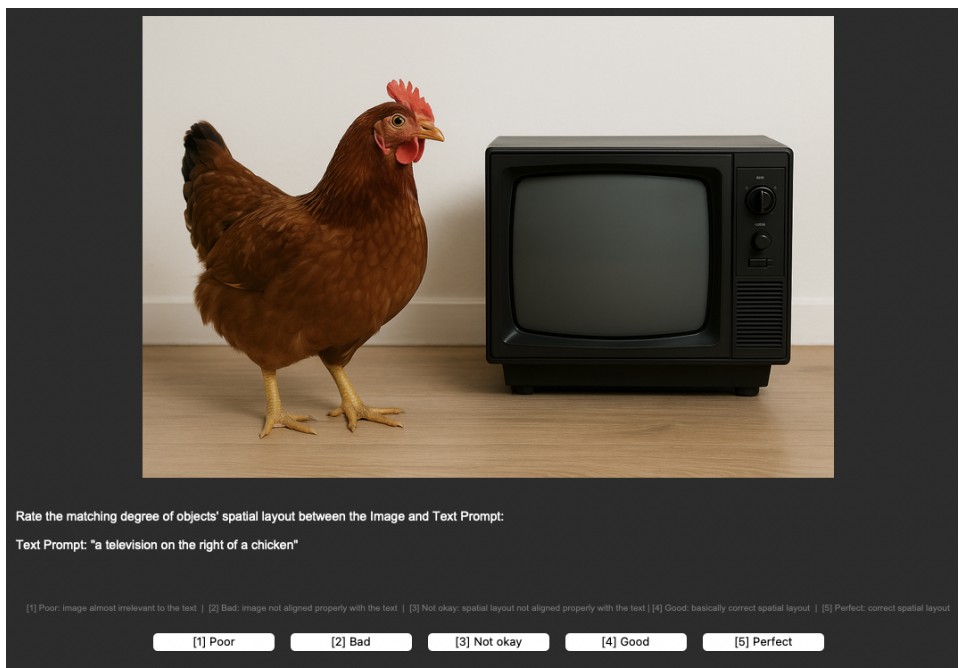

Figure 5: Human evaluation interface used to collect adherence responses for spatial dataset.

To validate the reliability of SANEval, we conducted a human evaluation study across all five categories of compositional adherence: spatial relationships, numeracy, and attribute binding (color, shape, and texture). For each case, annotators were presented with a generated image and the corresponding text prompt, and asked to judge whether the specified conditions were satisfied. The interface (see Figures 5–9) was designed to be uniform across tasks, with prompts and candidate images displayed together and response options provided in the form of Likert-style scales. This design ensured clarity and consistency, allowing workers to rate adherence from poor to perfect alignment while providing us with a direct basis for comparing human judgments with SANEval's automated scores.

# E    QUALITATIVE ANALYSIS

Table 5 provides illustrative examples from the SANEval benchmark, highlighting how scores are assigned and how diagnostic feedback pinpoints specific failure modes across different evaluation categories.

These examples demonstrate SANEval's ability to go beyond scalar scores by producing structured, interpretable feedback (e.g., reporting missing objects or incorrect attributes), which makes model shortcomings explicit and actionable.

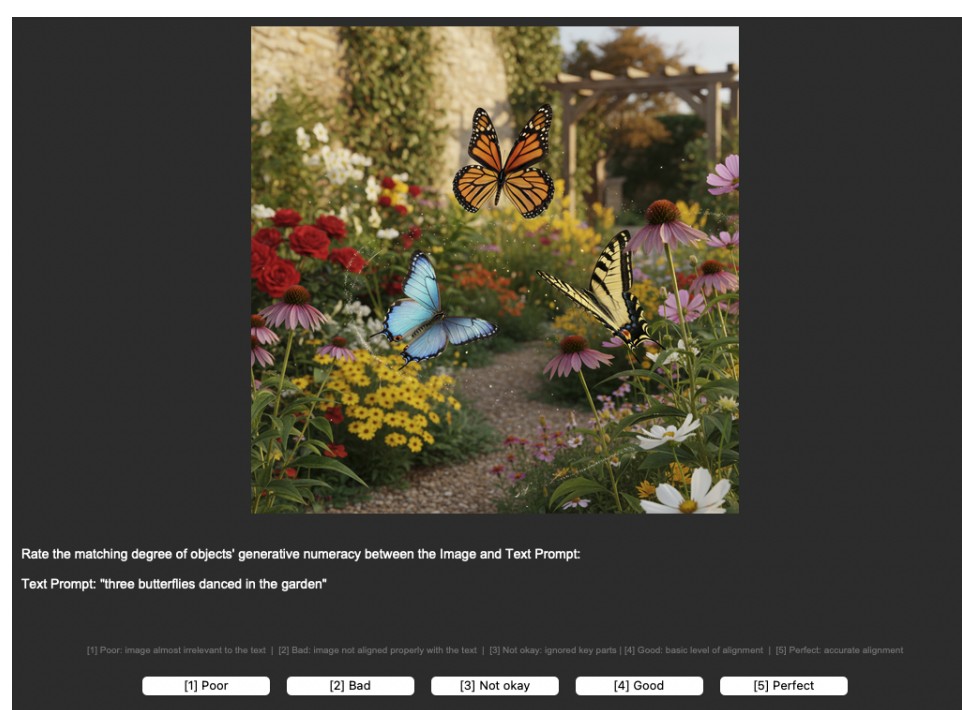

Figure 6: Human evaluation interface used to collect adherence responses for numeracy dataset.

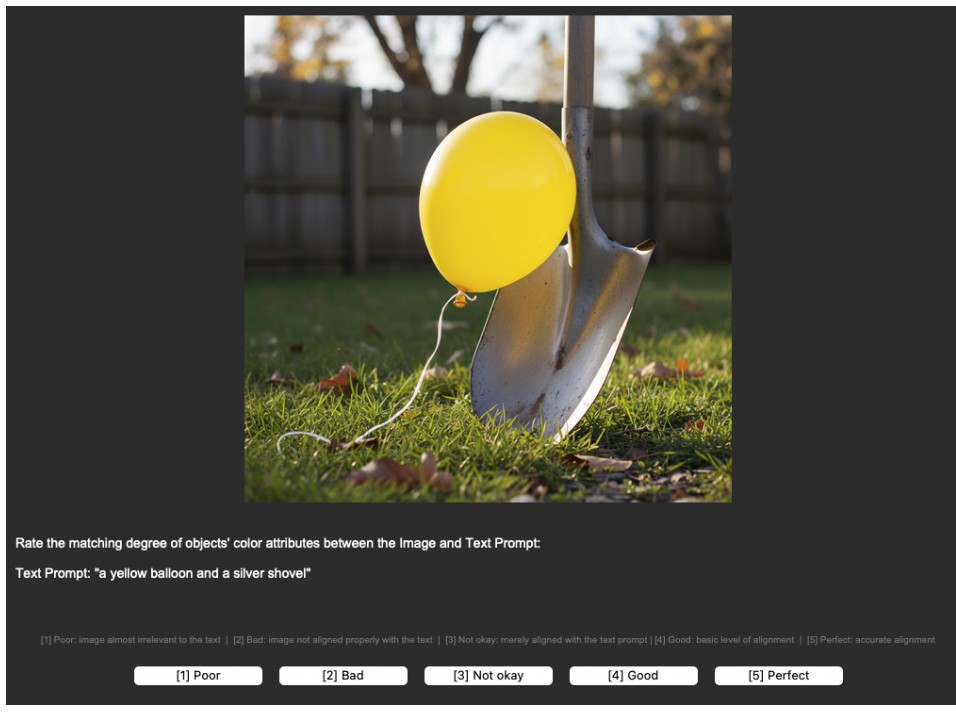

Figure 7: Human evaluation interface used to collect adherence responses for color binding dataset.

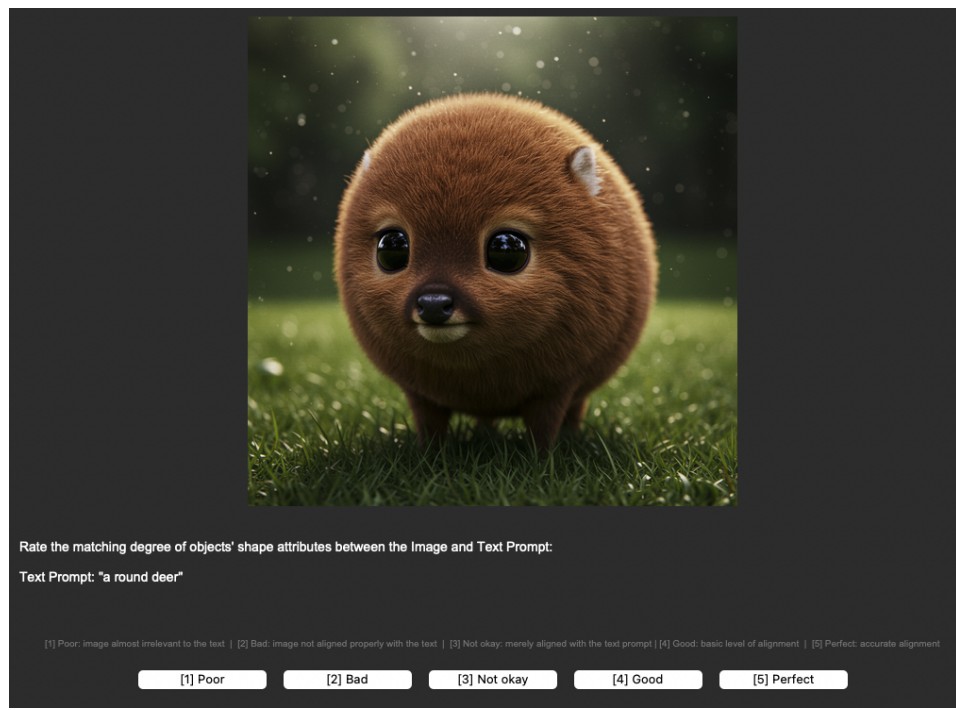

Figure 8: Human evaluation interface used to collect adherence responses for shape binding dataset.

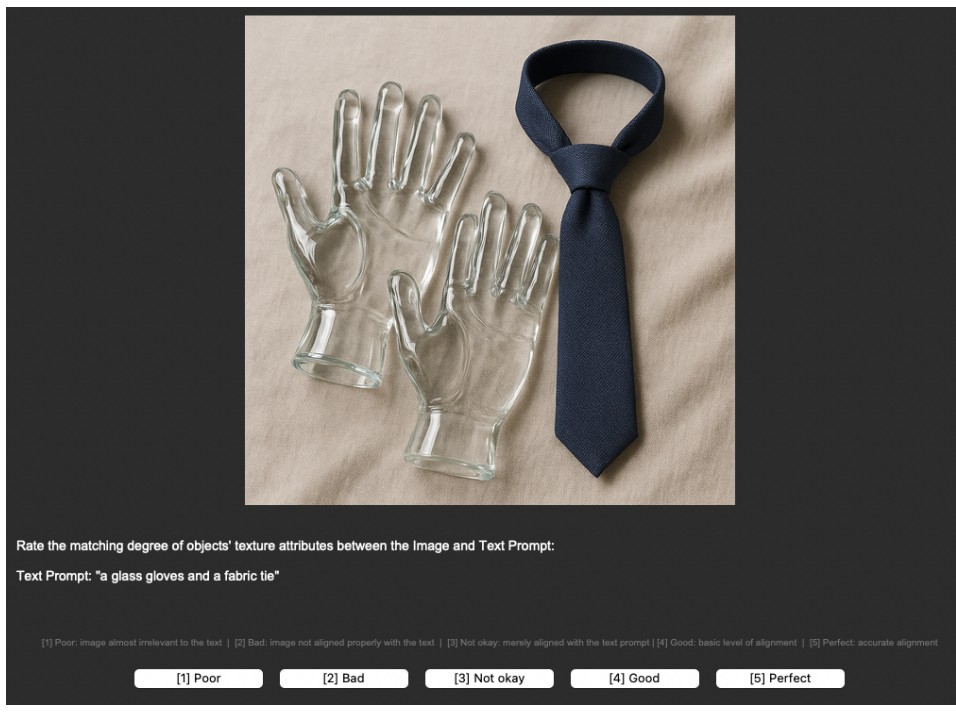

Figure 9: Human evaluation interface used to collect adherence responses for texture binding dataset.

Table 5: Qualitative sample prompts, generated images, scores, and diagnostic feedback from SANEval.

| Evaluation | Prompt | Image | Score | Conformity Feedback |
|---|---|---|---|---|
| Spatial | "a mouse on the bottom of a painting" |  | 0.0 | Missing object: painting |
| Numeracy | "four trucks" |  | 0.5 | Two trucks missing |
| Color Binding | "a brown shirt and a pink apple" |  | 0.2 | Missing object: shirt [brown]; Poor attribute binding for apple: expected [pink], score = 0.20 < 0.75 |
| Shape Binding | "a pentagonal spoon" |  | 0.0 | Poor attribute binding for spoon: expected [pentagonal], score = 0.00 < 0.75 |
| Texture Binding | "a rubber grass" |  | 0.3 | Poor attribute binding for grass: expected [rubber], score = 0.30 < 0.75 |

