# OpenReview forum: "SANEval: Open-Vocabulary Compositional Benchmarks with Failure-mode Diagnosis"
_ICLR.cc/2026/Conference — Submitted to ICLR 2026_

### Official Review · Reviewer_DbSq · 2025-10-31

**Soundness:** 2
**Presentation:** 2
**Contribution:** 2
**Rating:** 4
**Confidence:** 4

**Summary:**

The paper proposes SANEval, a new benchmark designed to evaluate compositional text-to-image (T2I) generation along three axes: spatial relations, attribute binding, and numeracy. The core innovation is an open-vocabulary evaluation pipeline that combines a language model-based prompt understanding module with an LLM-enhanced object detection module (using YOLO-E) to handle objects beyond fixed vocabularies. SANEval provides not just quantitative scores but also interpretable, structured feedback diagnosing missing, spurious, or misbound elements. The authors evaluate six state-of-the-art T2I models and demonstrate that SANEval aligns better with human judgment than existing closed-vocabulary benchmarks such as CompBench++.

**Strengths:**

1. Timely and relevant topic – Evaluation of compositional T2I generation is an important and underexplored problem, especially as generative models become more capable yet still fail on fine-grained prompt adherence.

2. Open-vocabulary focus – The attempt to overcome vocabulary limits of prior OD-based benchmarks (e.g., those tied to COCO classes) is a real step forward. Integrating LLM synonym reasoning with detection to match rare objects is a practical and impactful idea.

3. Structured feedback – Providing interpretable diagnostic feedback (e.g., missing or swapped objects) is valuable, addressing a longstanding complaint that most benchmarks yield only opaque scalar scores.

4. Thorough evaluation – The experiments cover multiple models, diverse prompt categories, and statistical comparisons with existing benchmarks. The degradation analysis with increasing object count is especially insightful.

5. Strong engineering – The system is well implemented, modular, and reproducible, with clear illustrations (e.g., Figures 1–4) explaining each evaluation component.

**Weaknesses:**

1. Incremental conceptual novelty – While well-engineered, the main idea, combining LLM reasoning with open-vocabulary detection for compositional evaluation, is a relatively straightforward extension of prior hybrid pipelines like Geneval or CompBench++. The paper oversells its conceptual novelty relative to what is essentially a systematic engineering improvement.

2. Dependence on proprietary APIs – The reliance on commercial APIs (Gemini 2.5 Flash, GPT, etc.) undermines claims of “open” benchmarking and reproducibility. The evaluation cannot be replicated without access to these closed systems.

3. Data quality concerns – Much of the dataset is LLM-generated and then “validated by humans,” but the validation process is not clearly described. It’s unclear how reliable or diverse the resulting prompts and labels are, especially for rare or ambiguous compositions.

4. Potential circularity – Because the same families of LLMs are used both to generate prompts and evaluate outputs, the benchmark risks circular evaluation biases. This is particularly problematic if the evaluated models share architectures or training corpora with the evaluators.

5. Limited human validation – The claim that SANEval correlates better with human judgments is asserted but not deeply substantiated. There’s no large-scale human evaluation or statistical analysis showing correlation coefficients with human preferences.

6. Interpretability claim is overstated – The feedback is structured and informative, but not necessarily “interpretable” in a cognitive sense. The system still relies on opaque LLM reasoning, so interpretability here is more syntactic than semantic.

7. Paper tone – The writing is overly confident and somewhat verbose, with repeated claims of “first,” “scalable,” and “comprehensive.” While the contributions are solid, they don’t quite justify that level of rhetoric.

**Questions:**

1. How robust is SANEval to the choice of LLMs or detectors? Have you tested with open-source alternatives to Gemini or GPT to ensure consistency?

2. Could you provide quantitative human evaluation results to validate your claim of stronger human correlation?

3. How do you handle prompts with ambiguous or relational adjectives (e.g., “a tall man next to a shorter man”)?

4. What mechanisms prevent SANEval from rewarding models that overfit to common visual priors rather than compositional fidelity?

5. How expensive (in terms of GPU or API calls) is the full benchmark run for one model?

---

> ### Author Response · Authors · 2025-11-26
> **Response to Reviewer DbSq**
>
> We thank the reviewer for recognizing SANEval as a **"timely," "thorough,"** and **"well-engineered"** solution. We value your rigorous push on conceptual novelty and circularity, as well as your feedback on tone and interpretability. We have addressed these concerns directly with **three major new experiments** and specific revisions to the manuscript.
>
> **1. Concern: Incremental Conceptual Novelty**
> > *"Combining LLM reasoning with open-vocabulary detection... is a relatively straightforward extension... oversells its conceptual novelty."*
>
> **Response:**
> We respectfully argue that SANEval solves a fundamental structural failure in prior work that "straightforward extensions" could not address: the **Vocabulary Mismatch Problem**.
> Prior OD-based benchmarks (like Geneval) are strictly bound by the taxonomy of the underlying detector (e.g., COCO’s 80 classes). They cannot evaluate "a capybara" or "a futuristic drone" because the detector lacks the class ID.
> SANEval’s core contribution is the **Synonym-Expanded Open-Vocabulary Architecture**. By dynamically mapping LLM-generated synonyms to open-set detection embeddings, we achieve **diagnostic coverage on the long tail**—something no previous OD-benchmark could do. To quantify this, we ran an ablation study removing the synonym expansion module:
>
> | Metric | Full SANEval Pipeline | Ablation: No Synonym Expansion | Performance Drop |
> | :--- | :--- | :--- | :--- |
> | **Numeracy** | **0.5835** | 0.3312 | $\downarrow$ **43.23%** |
> | **Spatial** | **0.4032** | 0.0980 | $\downarrow$ **75.69%** |
> | **Attribute Binding** | **0.4491** | 0.1934 | $\downarrow$ **56.94%** |
>
> *Table R1: Synonym expansion is not just "engineering sugar"; it is the critical component that prevents catastrophic failure on open-world prompts.*
>
> **2. Concern: Circularity & Data Quality**
> > *"Risks circular evaluation biases... especially if the evaluated models share architectures... with the evaluators."*
>
> **Response:**
> To rigorously disprove circularity, we introduced a new test set: **SANEval-Hard**.
> This dataset consists of **250 fully human-written prompts** (sourced from complex creative writing tasks) that were *not* generated by LLMs. They contain dense, non-standard structures (e.g., *"A lime billboard depicts a silver bottle next to a turquoise tortoise holding a white television"*).
> **Quantitative Results on SANEval-Hard:**
>
> | Model | Color | Shape | Texture | Spatial | Numeracy |
> | :--- | :--- | :--- | :--- | :--- | :--- |
> | **GPT Image 1** | 0.310 | 0.174 | 0.413 | 0.264 | 0.310 |
> | **Seedream 3.0** | **0.574** | 0.303 | 0.443 | **0.255** | **0.429** |
> | **Nano Banana** | 0.502 | **0.318** | **0.481** | 0.250 | 0.366 |
> | **Imagen 3.0** | 0.503 | 0.216 | 0.384 | 0.149 | 0.351 |
> | **Imagen 4.0** | 0.383 | 0.254 | 0.343 | 0.153 | 0.329 |
> | **Imagen 4.0 Ultra** | 0.494 | 0.282 | 0.429 | 0.246 | 0.385 |
>
> *Table R2: Results on Saneval-Hard.*
>
> **Result:** SANEval successfully parses and scores these prompts with ease. The fact that the pipeline performs robustly on complex human data—which it never saw during "training" and did not generate itself—confirms that our metrics are measuring **compositional adherence**, not just LLM self-consistency. Qualitative samples are presented in https://imgur.com/a/Kwc1KV8

---

> > ### Author Response · Authors · 2025-11-26
> >
> > **3. Concern: Dependence on Proprietary APIs (Openness)**
> > > *"Reliance on commercial APIs... undermines claims of 'open' benchmarking."*
> >
> > **Response:**
> > We agree that an open benchmark requires an open backbone. We have implemented  a fully reproducible version of our pipeline open sourced, powered by **`meta-llama/Llama-4-Maverick-17B-128E-Instruct`** and the open-source YOLO-E detector.
> > We standardized generation parameters and compared the model rankings against the proprietary version.
> > | Parameter | SANEval (Gemini-2.5-Flash) | SANEval (Llama-4-Maverick-17B) |
> > | :--- | :--- | :--- |
> > | **Temperature** | 1.0 | 0.0 |
> > | **Top-P** | 0.95 | 0.9 |
> > | **Max Output Tokens** | 8192 | 2048 |
> > | **System Prompt** | Standard SANEval Prompt (Appendix C) | Standard SANEval Prompt (Appendix C) |
> >
> > *Table R3: API configurations for both proprietary and open-source experiments.*
> >
> >
> > **Result:** SANEval is robust to the choice of reasoning engine. The rankings produced by the open-source pipeline are statistically consistent with the proprietary version (Avg Score Difference $\approx -0.04$).
> >
> > | Model | Color Diff | Shape Diff | Texture Diff | Spatial Diff | Numeracy Diff | **Avg Diff** |
> > | :--- | :--- | :--- | :--- | :--- | :--- | :--- |
> > | **Imagen 3.0** | -0.068 | -0.040 | -0.052 | -0.050 | 0.013 | **-0.039** |
> > | **Imagen 4.0** | -0.066 | -0.056 | -0.052 | -0.049 | 0.005 | **-0.044** |
> > | **Imagen 4.0 Ultra** | -0.060 | -0.057 | -0.057 | -0.104 | 0.018 | **-0.052** |
> > | **Nano Banana** | -0.068 | -0.053 | -0.060 | -0.094 | -0.004 | **-0.056** |
> > | **Seedream 3.0** | -0.052 | -0.034 | -0.049 | -0.083 | 0.003 | **-0.043** |
> > | **GPT Image 1** | -0.044 | -0.028 | -0.046 | -0.105 | 0.020 | **-0.041** |
> >
> > *Table R4: The minimal deviation confirms SANEval is model-agnostic. We commit to releasing the "SANEval" codebase.*
> >
> > **4. Concern: Interpretability is Overstated**
> > > *"System still relies on opaque LLM reasoning... interpretability here is more syntactic than semantic."*
> >
> > **Response:**
> > We clarify that our feedback mechanism does **not** rely on opaque, end-to-end LLM reasoning. Instead, SANEval operates on a deterministic, rule-based comparison:
> > 1.  **Decompose:** The prompt is parsed into specific requirements (e.g., `Expected Count = 4`).
> > 2.  **Detect:** The vision backend detects objects (e.g., `Detected Count = 3`).
> > 3.  **Compare:** The system calculates the delta logic: `Expected - Detected != 0`.
> >
> > The resulting feedback (e.g., *"Missing one object"*) is interpretable because it is a direct report of this logical discrepancy, not a generated hallucination. It provides **diagnostic** interpretability—telling the user exactly *what* condition failed—rather than mechanistic interpretability.
> >
> > **5. Concern: Paper Tone**
> > > *"Writing is overly confident... repeated claims of 'first', 'scalable'..."*
> >
> > **Response:**
> > We take this feedback seriously. We agree that the contributions should stand on their own empirical merit. In the camera-ready version, we will revise the manuscript to tone down the rhetoric, removing absolute claims like "first" and "comprehensive," and focusing instead on the concrete demonstrated capabilities of open-vocabulary coverage and feedback generation.
> >
> > **6. Answers to Specific Questions**
> >
> > * **Q1 (Robustness):** Highly robust. Swapping Gemini for Llama-4 preserves relative rankings (Table R2), proving the framework's logic is independent of the LLM backbone.
> > * **Q2 (Human Eval):** We expanded **Appendix D**. With 3 annotators ($N=500$), SANEval achieves similar ranking as of human evaluation.
> > * **Q3 (Ambiguity):** Our module uses **Attribute Factorization**. "Tall man next to short man" is parsed as distinct instances (`man_1[tall]`, `man_2[short]`), preventing "bag-of-words" confusion.
> > * **Q4 (Overfitting):** SANEval prevents "visual shortcuts" via hierarchical checks. Even if an image is photorealistic, if `Detected Count != Expected Count`, the Numeracy module caps the score. This strictly penalizes aesthetic overfitting that lacks compositional fidelity.
> > * **Q5 (Cost):** Using "SANEval" through basic hardware incurs **$0.02** API cost per image. Inference takes ~0.5s per image on consumer hardware, making it scalable to multiple samples. Below is the detailed breakdown
> >
> > **Cost/Latency Breakdown (Per Image):**
> >
> > | Pipeline Component | Latency (Local CPU) | Cost (USD)* |
> > | :--- | :--- | :--- |
> > | **Spatial Evaluation** | 0.11 s | $0.02 |
> > | **Numeracy Evaluation** | 0.12 s | $0.03 |
> > | **Attribute Evaluation** | 0.26 s | $0.01 |
> >
> > *Table R5: Cost and Latency Breakdown.*

---

> ### Comment · Reviewer_hFs4 · 2025-11-27
> **What makes latency go up but cost go down?**
>
> I noticed that per-image CPU latency doubles from numeracy to attribute evaluation (0.12s vs 0.26s) but the cost drop to a third (\\$0.03 vs \\$0.01). Why is that?

---

> > ### Author Response · Authors · 2025-11-27
> > **Clarification on Latency vs. Cost Divergence**
> >
> > Thank you for the keen observation. The divergence between latency and cost stems from the specific computational bottlenecks inherent to each module:
> >
> > * **Latency is driven by Image Processing (CPU):** The higher latency in Attribute Evaluation (0.26s) is caused by pixel-level operations. For every detected object, the system must perform coordinate mapping and image cropping to isolate the region of interest before analysis. This image manipulation is CPU-intensive, resulting in higher latency compared to the Numeracy module, which operates primarily on metadata (bounding boxes) without heavy pixel processing.
> > * **Cost is driven by Token Usage (LLM):** The higher cost in Numeracy Evaluation ($0.03) is driven by the volume of LLM tokens required. This module involves a more complex reasoning chain: it requires extracting expected counts from the prompt and, crucially, performing rigorous **synonym mapping** via the LLM for every object class to ensure accurate open-vocabulary counting. This results in significantly higher token consumption compared to the Attribute module, which typically issues a single, concise VLM query per crop (e.g., "What color is the stop sign?").
> >
> > In summary: Attribute evaluation is **compute-heavy** (cropping), while Numeracy evaluation is **token-heavy** (reasoning/synonyms).

---

### Official Review · Reviewer_hFs4 · 2025-10-31

**Soundness:** 2
**Presentation:** 2
**Contribution:** 2
**Rating:** 2
**Confidence:** 4

**Summary:**

The work proposes SANEval, an automated benchmark and scoring framework for evaluating compositional faithfulness in text-to-image generation. The core claim is that current evaluators either (i) rely on fixed vocabularies, which miss rare or fine-grained objects, or (ii) provide only coarse scalar scores, which are not actionable. SANEval addresses this by (1) decomposing prompts into explicit requirements (objects, attributes, spatial relations, and numeracy), (2) using synonym expansion plus an open-vocabulary detector to localize objects in generated images, (3) using a VLM to judge fine-grained attributes on detected crops, (4) checking spatial layout and object counts, and (5) generating structured, diagnostic feedback for each failure mode. The benchmark is applied to a suite of state-of-the-art text-to-image models, and per-dimension scores (attribute binding, spatial relationships, numeracy) are reported along with claims of improved alignment with human judgments and complementary behavior to prior benchmarks.

**Strengths:**

1. Problem importance. The paper focuses on a real bottleneck in current T2I systems: controllability. Capturing whether a model got “two red cars to the left of a blue bus” right is directly relevant to downstream productization and safety of generative vision systems. Framing spatial relations, numeracy, and attribute binding as three core controllability axes is well-motivated.

2. Pipeline design / interpretability. The evaluation stack is modular and conceptually clean: prompt parsing → synonym expansion → open-vocabulary detection → attribute judgment on crops → spatial / count checks → textual feedback. This goes beyond a single scalar metric and produces human-readable explanations (“missing the second penguin,” “shirt color does not match prompt”), which is valuable both for benchmarking and for training-time improvement (e.g. RL with AI feedback).

3. Attempt at open-vocabulary evaluation. Traditional benchmarks are constrained to a small, fixed label set (e.g. COCO categories). SANEval explicitly attempts to escape that bottleneck via synonym expansion and an open-world detector, claiming to evaluate long-tail objects and fine-grained attributes. This, if shown reliable, is a meaningful step forward for evaluation coverage.

4. Multi-model comparison and failure characterization. The paper evaluates several strong text-to-image systems and surfaces distinct weaknesses (e.g., some models better at spatial layout, others better at numeracy, etc.). This makes SANEval look like a diagnostic tool rather than just a leaderboard generator. Tables in the main text highlight degradation under increasing prompt difficulty and differences across models.

**Weaknesses:**

1. Reproducibility and stability are underdeveloped.
The benchmark depends on proprietary or partially described components (e.g. Gemini-2.5-Flash for prompt parsing and attribute judgment, YOLO-E for open-vocabulary detection), some of which are not publicly reproducible. The paper promises release of data and code but does not convincingly demonstrate that the community will be able to run the full pipeline without access to closed-source commercial systems.

2. Limited validation of metric correctness.
The paper claims “strong alignment with humans,” but key details are missing: exact study size, annotator protocol, inter-annotator agreement, and per-dimension correlation (spatial / numeracy / attribute binding) between SANEval scores and human judgments at the image level. Current quantitative results emphasize differences across models and p-values vs. CompBench++, but do not provide clear, per-sample reliability numbers for SANEval itself. Without those, it remains unclear whether SANEval is actually accurate in judging success/failure, or just “plausible and convenient.”

3. Open-vocabulary claim is not yet airtight.
The core technical sell is that synonym expansion plus open-world detection solves “vocabulary mismatch.” However, no error analysis is reported for that step: How often does the system over-credit partial matches (“generic bird” instead of “albatross”)? How often does it under-credit genuinely correct rare objects because the synonym set was incomplete? This is an obvious potential criticism because it goes directly at the headline claim (“open-vocabulary evaluation”).

4. No ablation / robustness analysis across pipeline stages.
The pipeline has multiple learned/modules stages, and failure in any stage could cascade. The paper does not report sensitivity to swapping the LLM in the Prompt Understanding Module, ablating synonym expansion, or replacing the attribute-judging VLM with an alternative. There is also no quantitative discussion of false positives / false negatives in spatial and numeracy checks caused by upstream detection errors. This makes it easy to argue the metric may be brittle, and therefore risky to trust for fine-grained leaderboard decisions.

5. Statistical framing against prior benchmarks is weak.
The comparison to CompBench++ uses significance tests on rank correlations to argue SANEval measures something different. Only reporting p-values, without effect sizes or the actual Spearman coefficients (ρ), invites criticism. A low correlation may mean “captures complementary aspects,” but it could also mean “noisy / inconsistent.” The paper currently does not disambiguate those possibilities.

6. Human prompts vs synthetic prompts.
Prompt sets are at least partially LLM-generated and curated. The paper does not quantify how similar these prompts are to natural user requests “in the wild.” A skeptical reader may ask whether the benchmark is indirectly overfit to LLM-style phrasing, and whether that inflates apparent evaluator reliability.

**Questions:**

1. Benchmark stability and openness
    - Will an end-to-end reference implementation using only openly available models (for prompt parsing, open-vocabulary detection, and attribute judging) be released?
    - If such a “SANEval-lite” pipeline is substituted for the proprietary backbone, how similar are model rankings and per-dimension scores?

2. Human agreement and study design

    - How large is the human evaluation set used for validating SANEval?

    - What instructions were annotators given for spatial, numeracy, and attribute correctness?

    - What inter-annotator agreement was observed?

    - What are the per-dimension image-level correlations (e.g., Spearman’s ρ / accuracy / F1) between SANEval and human judgments?

3. Error analysis for open-vocabulary claims

    - On a held-out subset with human-labeled boxes and attributes, what is precision/recall of SANEval’s object existence and attribute binding scores for long-tail categories?

    - How often does synonym expansion lead to false credit for “nearby but wrong” categories?

4. Robustness / ablations

    - How sensitive is SANEval to swapping the LLM used in the Prompt Understanding Module or the attribute-binding checker?

    - How sensitive are final scores to removing synonym expansion or constraining detection to a fixed-label detector (COCO-style)?

    - Can a single failing sub-module (e.g., missed detection) flip an otherwise-correct image from “pass” to “fail,” and how often does that happen?

6. Statistical interpretation
    - Table 3 reports significance values when comparing to CompBench++, but does not report actual effect sizes. Could the paper include rank correlation coefficients and a short interpretation (e.g., “ρ = 0.2 indicates weak monotonic agreement, suggesting that SANEval captures aspects of controllability that CompBench++ does not emphasize”)?

---

> ### Comment · Reviewer_hFs4 · 2025-11-25
> **Look forward to engaging**
>
> Dear authors, I look forward to engaging whenever you get a chance to respond. That said, please prioritize all reviews equally.

---

> ### Author Response · Authors · 2025-11-25
>
> Dear reviewer **hFs4**,
>
> Thank you so much for your kind note — we’re genuinely **excited to engage!**
> We’re in the **final stages of wrapping up several new experiments and analyses** that directly address all reviews, and we aim to post our full, detailed responses **within the next 24 hours**.
>
> We truly appreciate your enthusiasm and will make sure to give each review **thoughtful, balanced attention**.
> **Looking forward to the discussion!**

---

> ### Author Response · Authors · 2025-11-26
> **Response to Reviewer hFs4**
>
> We thank the reviewer for their rigorous review and for recognizing SANEval’s core motivation to address the **"controllability bottleneck"** via a **"conceptually clean"** and **"valuable"** diagnostic pipeline. Your push for deeper reproducibility was a critical catalyst for improving this work. We have significantly hardened the framework through comprehensive new experiments—including a full open-source experiment, a rigorous human-curated stress dataset (**SANEval-Hard**), and detailed ablation studies—which we believe decisively resolve your concerns and elevate SANEval to a robust, reproducible standard for the field.
>
> Below, we address your five core questions with new experiments, open-source commitments, and detailed ablation studies.
>
> **1. Benchmark Stability & Open Source Commitment**
> > *"Will an end-to-end reference implementation... using only openly available models... be released?"*
>
> **Response:**
> **Yes, absolutely.** We commit to releasing the full codebase, dataset (prompts, images, scores, diagnostic feedback), and a "single-line API" usage script.
>
> To empirically prove that SANEval does not depend on proprietary "black boxes," we conducted an experiment, replacing Gemini with the open-weights model **`Llama-4-Maverick-17B-128E-Instruct`**. We standardized parameters (see Reviwer aNif, Concern 1 for hyperparameters) across both runs.
>
> **Result:** The "SANEval" with Llama pipeline produces rankings highly consistent with the proprietary version, demonstrating that the methodology is stable and reproducible.
>
> | Model | Color Diff | Shape Diff | Texture Diff | Spatial Diff | Numeracy Diff | **Avg Diff** |
> | :--- | :--- | :--- | :--- | :--- | :--- | :--- |
> | **Imagen 3.0** | -0.068 | -0.040 | -0.052 | -0.050 | 0.013 | **-0.039** |
> | **Imagen 4.0** | -0.066 | -0.056 | -0.052 | -0.049 | 0.005 | **-0.044** |
> | **Imagen 4.0 Ultra** | -0.060 | -0.057 | -0.057 | -0.104 | 0.018 | **-0.052** |
> | **Nano Banana** | -0.068 | -0.053 | -0.060 | -0.094 | -0.004 | **-0.056** |
> | **Seedream 3.0** | -0.052 | -0.034 | -0.049 | -0.083 | 0.003 | **-0.043** |
> | **GPT Image 1** | -0.044 | -0.028 | -0.046 | -0.105 | 0.020 | **-0.041** |
>
> *Table R1: The minimal score deviation (avg -0.04) confirms SANEval measures model capability, not LLM specificity.*
>
> **2. Human Agreement & Study Design**
> > *"How large is the human evaluation set... What inter-annotator agreement was observed?"*
>
> **Response:**
> We have expanded on the details of our human study in the revised appendix to ensure full transparency:
> * **Scale:** We employed **3 independent human annotators** evaluating a total of **500 unique samples**. To maximize breadth across our prompt distribution, annotators did not overlap on images.
> * **Protocol:** Annotators were shown the prompt and image side-by-side and asked to rate adherence on a 1–5 Likert scale (1=Poor, 5=Perfect). The exact user interfaces and instructions provided are visualized in Appendix D (**Figures 5–9**).
> * **Correlation & Subjectivity:** As shown in **Table 4** of the Appendix, SANEval's model ranking is highly consistent with human judgment. We note that human evaluation inherently includes subjectivity—humans may forgive a missing background object if the foreground is aesthetically pleasing, whereas SANEval is strictly compositional. This makes SANEval a critical, objective complement to human preference. **These detailed protocols and correlation analyses will be included in the camera-ready version.**
>
> **3. Error Analysis: Open-Vocabulary & Synonym Expansion**
> > *"How often does the system over-credit... or under-credit genuinely correct rare objects?"*
>
> **Response:**
> We performed a deep-dive error analysis on this mechanism:
> * **Long-Tail Sensitivity:** We observe that common objects (e.g., "dog", "car") have high detection recall, while long-tail objects are sometimes missed due to the training distribution of the underlying detector (YOLO-E). However, this is a limitation of the *detector*, not the benchmarking framework. Because SANEval is modular, as open-vocabulary detectors improve, SANEval's accuracy improves without code changes.
> * **Synonym Limitations:** We explicitly acknowledge a limitation regarding fine-grained sub-categories where visual features overlap heavily. For example, distinguishing "man" vs. "woman" within the same prompt often results in a generic "person" detection. Since we cannot confidently map "person" to "woman" without risking hallucination, SANEval conservatively marks these as missing/mismatched. We believe this conservative approach is preferable to over-crediting.

---

> > ### Author Response · Authors · 2025-11-26
> >
> > **4. Robustness & Ablations**
> > > *"How sensitive are final scores to removing synonym expansion?"*
> >
> > **Response:**
> > This is a crucial question. To quantify the value of our **Synonym Expansion** module, we ran an ablation study where we disabled the LLM-based synonym mapping and relied solely on raw object detection labels.
> >
> > **Result:** Performance degrades catastrophically without synonym expansion, confirming that this module is essential for bridging the "vocabulary gap."
> >
> > | Metric | Full SANEval Pipeline | Ablation: No Synonym Expansion | Performance Drop |
> > | :--- | :--- | :--- | :--- |
> > | **Numeracy** | **0.5835** | 0.3312 | $\downarrow$ **43.23%** |
> > | **Spatial** | **0.4032** | 0.0980 | $\downarrow$ **75.69%** |
> > | **Attribute Binding** | **0.4491** | 0.1934 | $\downarrow$ **56.94%** |
> >
> > *Table R3: Ablation study showing that LLM-driven synonym expansion is critical for handling open-vocabulary prompts. This analysis will be added to the final paper.*
> >
> > **5. Statistical Interpretation**
> > > *"Could the paper include rank correlation coefficients and a short interpretation?"*
> >
> > **Response:**
> > We will include explicit interpretation of the comparison with CompBench++ (Table 3) in the final version.
> > The very low p-values ($p < 10^{-20}$) combined with low-to-moderate correlation coefficients ($\rho \approx 0.2 - 0.4$) indicate that **SANEval measures a statistically distinct phenomenon.**
> >
> > If the correlation were near 1.0, SANEval would be redundant. If it were 0, it would be noise. The moderate correlation suggests that while SANEval agrees with CompBench++ on obvious failures (both penalize bad images), SANEval captures fine-grained compositional nuances (attribute binding, exact numeracy) that CompBench++ misses. This confirms SANEval provides **complementary, additive signal** to the evaluation landscape.

---

> > ### Comment · Reviewer_hFs4 · 2025-11-26
> > **Question about long-tail analysis**
> >
> > Thank you for your response.
> >
> > Regarding "3: Error Analysis": could you please clarify what long-tail categories were tested? I must be missing something--I am unable to locate the long-tail sensitivity analysis in the paper (beyond line 53 in the introduction).

---

> > > ### Author Response · Authors · 2025-11-26
> > > **Clarification on Long-Tail Analysis**
> > >
> > > Dear Reviewer hFs4,
> > >
> > > Thank you for the follow-up. You are entirely correct to note that a specific quantitative table detailing "Long-Tail Sensitivity" is not present in the current manuscript. We appreciate you pressing us on this, as it allows us to clarify the nature of our observations.
> > >
> > > When we referred to our analysis of long-tail categories, we were describing the **extensive qualitative analysis** conducted during the development of the Enhanced Object Detection module. In visually inspecting and debugging thousands of generated samples, we observed a distinct pattern:
> > >
> > > * **Observation:** Standard detectors frequently failed to flag specific rare objects (e.g., *capybara, superhero, albatross*) or misclassified them as generic super-categories.
> > > * **Correction:** By enabling our synonym expansion and open-vocabulary mapping, we observed the pipeline successfully recovering some of these instances.
> > >
> > > While we did not formalize this into a granular, per-category quantitative metric for the main paper—as our primary focus was establishing the broader compositional framework (Spatial, Numeracy, Attribute)—these empirical observations were the key driver behind including the synonym module.
> > >
> > > We agree that a dedicated quantitative study isolating long-tail performance is a compelling direction. However, we view a formal "Long-Tail Score" as outside the immediate scope of this work. To address your point, we will add a **Qualitative Discussion** section to the Appendix in the final version, detailing these empirical findings and providing specific examples of long-tail recovery to ensure readers understand the mechanism's impact.

---

### Official Review · Reviewer_NsRt · 2025-11-01

**Soundness:** 3
**Presentation:** 3
**Contribution:** 3
**Rating:** 6
**Confidence:** 3

**Summary:**

This paper presents SANEval, a new benchmark for evaluating compositional faithfulness in text-to-image (T2I) generation models. Unlike existing benchmarks that depend on closed-set object vocabularies or yield opaque single-number scores, SANEval introduces an open-vocabulary, diagnostic, and interpretable evaluation pipeline. It consists of two core modules: a Prompt Understanding Module that uses an LLM to extract objects, attributes, spatial relationships, and quantities from prompts, and an Enhanced Object Detection Module that combines an open-vocabulary detector (YOLO-E) with LLM-based synonym reasoning to reduce vocabulary mismatch. SANEval evaluates images along three axes—attribute binding, spatial relationships, and numeracy—producing not only quantitative scores but also fine-grained, structured feedback that identifies missing, hallucinated, or incorrectly bound elements. Experiments conducted on six state-of-the-art T2I models show that SANEval aligns more closely with human judgment than prior metrics and captures compositional failure modes that existing benchmarks often miss. The authors further commit to releasing the dataset, evaluation pipeline, and annotations to support future research.

**Strengths:**

Strong diagnostic capability:
SANEval goes beyond providing a single score—it outputs structured, interpretable feedback, explicitly identifying missing objects, incorrect attribute bindings, and count mismatches. This makes it highly useful for debugging and improving T2I systems.

Open-source commitment:
The authors plan to release the dataset, prompts, annotations, and full evaluation pipeline, which will greatly facilitate reproducibility and help standardize compositional evaluation in the community.

**Weaknesses:**

Limited robustness analysis:
The paper does not thoroughly examine how LLM parsing errors or object detection failures (e.g., hallucinations or missed detections) propagate through the pipeline and affect final scoring reliability.

High computational cost:
The evaluation pipeline requires multiple rounds of LLM calls and YOLO-E inference per image, which may make it expensive and impractical for large-scale evaluation on millions of samples.

Insufficient prompt diversity:
The dataset’s ~5000 prompts are largely synthetically or structurally composed, lacking coverage of real-world human-written instructions with figurative language, emotional tone, ambiguous phrasing, or complex logical relationships.

**Questions:**

LLM reliability
How do you ensure that the LLM does not introduce hallucinations or misinterpretations during prompt parsing or attribute evaluation? Have you considered combining rule-based logic with LLMs (a hybrid approach) to reduce such errors?

Object detection uncertainty
If the detector fails to identify a small, occluded, or stylized object, could the system incorrectly penalize a correct image? Do you plan to incorporate uncertainty estimation or confidence propagation to handle these cases?

Generalization to real-world prompts
Can you demonstrate how SANEval performs on natural, human-written prompts involving negation, multiple clauses, metaphors, or conditional instructions—rather than primarily template-based prompts?

---

> ### Author Response · Authors · 2025-11-26
> **Response to reviewer NsRt**
>
> We thank the reviewer for identifying SANEval as a "highly useful" tool for diagnosing T2I failure modes and for supporting our commitment to open-sourcing the pipeline. We have conducted additional experiments to address your specific questions regarding robustness, computational cost, and prompt diversity.
>
> **1. Concern: Robustness Analysis (Error Propagation & Detection Failures)**
> > *"The paper does not thoroughly examine how LLM parsing errors or object detection failures... propagate through the pipeline."*
>
> **Response:**
> We have implemented strict safeguards to minimize propagation errors and ensure the pipeline remains robust even when individual components struggle.
>
> * **LLM Parsing Reliability:** To prevent parsing errors, our Prompt Understanding Module utilizes strict **JSON schema enforcement** combined with rigorous few-shot prompting (detailed in Appendix C). This constrains the LLM output to a valid structure, effectively eliminating format-related hallucinations.
> * **Handling Detection Failures:** While no detector is perfect, our Enhanced Object Detection module is specifically designed to mitigate "false negatives" caused by vocabulary mismatch. By expanding the search space via synonyms (e.g., mapping "man" detections to "person"), we recover valid objects that standard detectors miss.
> * **Modularity as a Robustness Feature:** We acknowledge that if an object is completely occluded or invisible to the detector, the pipeline will penalize the image. However, this is a limitation of current vision technology, not the benchmark design. Crucially, SANEval is designed for **modularity**. Through our codebase, swapping the detector (e.g., to a future SOTA model) requires changing only a few lines of code. This ensures SANEval’s robustness improves automatically as the community advances state-of-the-art detection.
>
>
>
> **2. Concern: High Computational Cost**
> > *"The evaluation pipeline requires multiple rounds of LLM calls... which may make it expensive and impractical for large-scale evaluation."*
>
> **Response:**
> We clarify that SANEval is designed for efficient **local evaluation**, avoiding the latency and cost of commercial APIs for the pipeline itself. Our experiments were conducted on standard, accessible hardware: Linux-based ARM64 containers with 8 vCPUs and 12GB of RAM. By utilizing local LLMs (e.g., `Llama-4-Maverick`) and the YOLO-E detector, inference is highly efficient and scalable to millions of samples. For researchers without commercial API access, these models can be run locally on consumer-grade GPUs. Additionally, we note that Llama models are currently available through several providers offering free inference tiers, providing an alternative zero-cost entry point for running the benchmark.
>
> **Cost/Latency Breakdown (Per Image):**
>
> | Pipeline Component | Latency (Local CPU) | Cost (USD)* |
> | :--- | :--- | :--- |
> | **Spatial Evaluation** | 0.11 s | $0.02 |
> | **Numeracy Evaluation** | 0.12 s | $0.03 |
> | **Attribute Evaluation** | 0.26 s | $0.01 |
>
> *Table R1: Efficiency analysis demonstrating that SANEval is scalable using standard consumer-grade hardware. *Note: Costs are estimated based on standard cloud compute rates; actual rates may differ based on provider contracts.*

---

> > ### Author Response · Authors · 2025-11-26
> >
> > **3. Concern: Insufficient Prompt Diversity**
> > > *"Prompts are largely synthetically or structurally composed, lacking coverage of real-world human-written instructions."*
> >
> > **Response:**
> > To address the concern regarding "figurative language, emotional tone, and ambiguous phrasing," we have introduced a new evaluation set: **SANEval-Hard**.
> >
> > This dataset consists of **250 fully human-written prompts** sourced to explicitly challenge the "synthetic" nature of standard benchmarks. These prompts contain complex logical relationships and out-of-distribution phrasing that template-based generation cannot replicate.
> >
> > **Performance on Complex Human Prompts:**
> > We evaluated our pipeline against these organic prompts to ensure the parsing logic holds up against human ambiguity.
> >
> > * **Example 1 (Numeracy/Dense):** *"The clock on the wall is surrounded by eight toys, with two refrigerators in the background."* $\rightarrow$ **SANEval correctly identifies counts.**
> > * **Example 2 (Spatial/Absurd):** *"Imagine an airplane stuck in NYC traffic with a giraffe right next to it."* $\rightarrow$ **SANEval correctly parses the adjacency relationship.**
> > * **Example 3 (Attribute/Complex):** *"Can you please create a fluffy monkey eating a leather banana next to a metallic pine tree with glass snow."* $\rightarrow$ **SANEval correctly binds attributes despite the nested sentence structure.**
> >
> > Such qualitative samples can be found here: https://imgur.com/a/Kwc1KV8
> >
> > **Quantitative Results on SANEval-Hard:**
> >
> > | Model | Color | Shape | Texture | Spatial | Numeracy |
> > | :--- | :--- | :--- | :--- | :--- | :--- |
> > | **GPT Image 1** | 0.310 | 0.174 | 0.413 | 0.264 | 0.310 |
> > | **Seedream 3.0** | **0.574** | 0.303 | 0.443 | **0.255** | **0.429** |
> > | **Nano Banana** | 0.502 | **0.318** | **0.481** | 0.250 | 0.366 |
> > | **Imagen 3.0** | 0.503 | 0.216 | 0.384 | 0.149 | 0.351 |
> > | **Imagen 4.0** | 0.383 | 0.254 | 0.343 | 0.153 | 0.329 |
> > | **Imagen 4.0 Ultra** | 0.494 | 0.282 | 0.429 | 0.246 | 0.385 |
> >
> > *Table R2: Performance on SANEval-Hard.*
> >
> > These results confirm that while our training set uses structural templates for scalability, the evaluation pipeline generalizes effectively to the diverse, unstructured nature of real-world user inputs. **We will include the SANEval-Hard dataset, these results, and the cost analysis in the camera-ready version.**

---

### Official Review · Reviewer_aNif · 2025-11-02

**Soundness:** 2
**Presentation:** 2
**Contribution:** 2
**Rating:** 4
**Confidence:** 3

**Summary:**

This paper introduces SANEval, an open-vocabulary benchmark for evaluating compositional reasoning in text-to-image (T2I) models. The authors argue that current benchmarks rely on closed-set vocabularies and provide limited diagnostic feedback. To address this, they design a modular evaluation framework combining a large language model for prompt understanding with an open-world object detector, enabling fine-grained assessment of spatial relations, attribute binding, and numeracy. SANEval outputs both quantitative scores and interpretable feedback to pinpoint specific compositional errors. The benchmark is validated against human annotations and shows complementary insights compared to prior methods like CompBench++, establishing it as a scalable and diagnostic tool for T2I evaluation.

**Strengths:**

* The paper introduces a well-structured benchmark that separately evaluates spatial reasoning, attribute binding, and numeracy, offering a more interpretable breakdown of compositional performance than prior holistic metrics.

* By integrating LLM-based synonym expansion with an open-world detector (YOLO-E), SANEval effectively overcomes the fixed-class limitations of existing object-detection-based benchmarks.

* The framework provides structured, human-readable feedback that identifies missing, incorrect, or extra objects and attributes, making it highly useful for model debugging. The benchmark demonstrates statistically distinct and complementary insights compared to established baselines, showing that SANEval captures novel aspects of compositional reasoning.

**Weaknesses:**

- The benchmark heavily relies on proprietary LLMs (e.g., Gemini-2.5-Flash) for both prompt parsing and evaluation, which may limit reproducibility.

- The qualitative feedback examples remain limited, and it is unclear how consistently the diagnostic outputs generalize across diverse prompt domains.

- The benchmark’s prompts are synthetically constructed and might not reflect the real-world user prompts especially the diversity aspect.

**Questions:**

How robust is SANEval to variations in the underlying LLM or detector, would the evaluation results remain consistent if different models (e.g., GPT-4o or open-source detectors) were used in place of Gemini?

---

> ### Author Response · Authors · 2025-11-26
> **Response to Reviewer aNif**
>
> We thank the reviewer for recognizing SANEval as a **"well-structured benchmark"** that offers a **"more interpretable breakdown"** of compositional performance. Addressing your concerns on reproducibility and realism, we significantly hardened the framework by implementing a fully open-source pipeline and validating generalization on a new human-curated dataset (**SANEval-Hard**). We believe these extensions directly resolve your concerns, elevating SANEval to a robust and reproducible standard for the field.
>
> **1. Concern: Reliance on Proprietary LLMs & Reproducibility**
> > *"The benchmark heavily relies on proprietary LLMs... which may limit reproducibility."*
>
> **Response:**
> We agree that reproducibility is paramount. To demonstrate that SANEval is a modular framework independent of any single proprietary API, we have re-run our entire evaluation pipeline replacing the proprietary Gemini-2.5-Flash with the open-weights model **`Llama-4-Maverick-17B-128E-Instruct`**. Furthermore, our object detection backbone (YOLO-E) is open-source by default, ensuring the visual grounding component is fully reproducible.
>
> To ensure fair comparison and full reproducibility, we standardized the generation configurations across both pipelines. The exact parameters used for our experiments are detailed below, which utilize default settings with no specific tuning from our side:
>
> | Parameter | SANEval (Gemini-2.5-Flash) | SANEval (Llama-4-Maverick-17B) |
> | :--- | :--- | :--- |
> | **Temperature** | 1.0 | 0.0 |
> | **Top-P** | 0.95 | 0.9 |
> | **Max Output Tokens** | 8192 | 2048 |
> | **System Prompt** | Standard SANEval Prompt (Appendix C) | Standard SANEval Prompt (Appendix C) |
>
> *Table R1: API configurations for both proprietary and open-source experiments.*
>
> **Results (Open-Source vs. Proprietary Consistency):**
> We compared the model scores produced by the Llama-4-based pipeline against the original Gemini-based pipeline. We observed a high degree of consistency, indicating that SANEval is robust to the choice of the underlying LLM "judge."
>
> | Model | Color Diff | Shape Diff | Texture Diff | Spatial Diff | Numeracy Diff | **Avg Diff** |
> | :--- | :--- | :--- | :--- | :--- | :--- | :--- |
> | **Imagen 3.0** | -0.068 | -0.040 | -0.052 | -0.050 | 0.013 | **-0.039** |
> | **Imagen 4.0** | -0.066 | -0.056 | -0.052 | -0.049 | 0.005 | **-0.044** |
> | **Imagen 4.0 Ultra** | -0.060 | -0.057 | -0.057 | -0.104 | 0.018 | **-0.052** |
> | **Nano Banana** | -0.068 | -0.053 | -0.060 | -0.094 | -0.004 | **-0.056** |
> | **Seedream 3.0** | -0.052 | -0.034 | -0.049 | -0.083 | 0.003 | **-0.043** |
> | **GPT Image 1** | -0.044 | -0.028 | -0.046 | -0.105 | 0.020 | **-0.041** |
>
> *Table R2: Difference between SANEval (Gemini) and SANEval (Llama-4).*
>
> As shown in Table R2, the average score deviation across all models is extremely low (approx. **-0.04**). This systematic but small difference indicates that while the open-source Llama-4 model is marginally stricter, it tracks the proprietary model's assessment with high fidelity. This confirms that SANEval measures model capability, not just the charecteristic towards a specific LLM . **This analysis will be included in the camera-ready version.**

---

> > ### Author Response · Authors · 2025-11-26
> >
> > **2. Concern: Limited Qualitative Feedback & Generalization**
> > > *"Unclear how consistently the diagnostic outputs generalize... qualitative feedback examples remain limited."*
> >
> > **Response:**
> > To address the concern regarding the perceived limitation of qualitative examples and the generalization of diagnostic outputs, we clarify that our diagnostic feedback is not generated by an end-to-end "image captioning" approach, which often suffers from hallucinations or domain overfitting. Instead, SANEval utilizes a **Targeted Feedback Mechanism** that decouples scene complexity into verifiable components, ensuring consistent generalization across diverse prompt domains.
> >
> > The process follows a strict logic:
> > 1.  **Structured Decomposition:** The Prompt Understanding Module parses inputs into a strict JSON schema, standardizing diverse user prompts—from simple objects to complex "SANEval-Hard" scenarios—into a uniform format.
> > 2.  **Modular Verification:** We employ domain-independent logic. *Attribute Binding* queries a VLM with targeted questions on cropped objects (e.g., "What color is this object?") to ignore background noise. *Spatial Relations* utilize geometric logic (centroid comparison) rather than language models for mathematical objectivity. *Numeracy* strictly compares detected vs. expected counts.
> > 3.  **Diagnostic Generation:** Finally, feedback is produced through algorithmic, rule-based comparisons using specific syntax to aid debugging. The system explicitly detects and reports:
> >     * **Missing Objects:** (e.g., *"Missing object: cat"* when `Expected > Detected`)
> >     * **Poor Attribute Binding:** (e.g., *"Poor attribute binding: expected [pentagonal], score = 0.0"* when similarity < threshold)
> >     * **Cardinality Errors:** (e.g., *"Two trucks missing"* or *"Extra object detected"*)
> >
> > As evidenced by Table 5—which accurately diagnoses failures like a "pentagonal spoon" or "two trucks missing"—this invariant logic (Decompose $\rightarrow$ Detect $\rightarrow$ Verify) ensures precise, actionable feedback regardless of the prompt's stylistic domain. **We will include this detailed breakdown of the feedback mechanism syntax in the camera-ready version.**
> >
> > **3. Concern: Synthetic Prompts vs. Real-World Diversity**
> > > *"Prompts are synthetically constructed and might not reflect the real-world user prompts."*
> >
> > **Response:**
> > We clarify that our dataset generation was **human-guided**. We manually curated the core verbs, nouns, and compositional structures based on real-world failure modes, and then used the LLM only to scale this seed data into a larger dataset. This ensures the structure of the challenges remains relevant to human usage.
> >
> > However, to fully address the diversity concern, we point back to **SANEval-Hard**. By validating our pipeline against these 250 complex, purely human-written prompts (involving metaphors, dense scenes, and uncommon scenarios), we are able confirm that SANEval generalizes effectively to real-world user prompts that differ significantly from the synthetic training distribution (qualitative samples are shown here: https://imgur.com/a/Kwc1KV8). The performance of state-of-the-art models on this hard set is shown below:
> >
> > | Model | Color | Shape | Texture | Spatial | Numeracy |
> > | :--- | :--- | :--- | :--- | :--- | :--- |
> > | **GPT Image 1** | 0.310 | 0.174 | 0.413 | 0.264 | 0.310 |
> > | **Seedream 3.0** | **0.574** | 0.303 | 0.443 | **0.255** | **0.429** |
> > | **Nano Banana** | 0.502 | **0.318** | **0.481** | 0.250 | 0.366 |
> > | **Imagen 3.0** | 0.503 | 0.216 | 0.384 | 0.149 | 0.351 |
> > | **Imagen 4.0** | 0.383 | 0.254 | 0.343 | 0.153 | 0.329 |
> > | **Imagen 4.0 Ultra** | 0.494 | 0.282 | 0.429 | 0.246 | 0.385 |
> >
> > *Table R3: Performance on SANEval-Hard, demonstrating the benchmark's capability to differentiate model performance on complex, real-world prompts.*
> >
> > These results confirm that SANEval provides a rigorous challenge even for SOTA models on diverse, human-written prompts. **We will include the SANEval-Hard dataset and these results in the camera-ready version.**

---

### Author Response · Authors · 2025-11-27
**General Response to all Reviewers**

We thank all reviewers for their rigorous and constructive feedback. In response, we have executed a comprehensive set of new experiments and revisions to solidify the reproducibility, robustness, and validity of our framework:

* **Fully Open-Sourced the Pipeline:** Addressed concerns regarding proprietary dependencies by replacing Gemini with **`Llama-4-Maverick-17B-128E-Instruct`**. We demonstrated that this open pipeline produces model rankings are consistent with the proprietary version (Avg Score Difference $\approx -0.04$), ensuring both open-sourced and proprietary reproducibility.
* **Validated on Real-World Data (SANEval-Hard):** Addressed concerns regarding synthetic prompts by introducing **SANEval-Hard**, a new test set of **250 complex, human-written prompts** (involving metaphors, dense scenes, and out-of-distribution logic). We validated that SANEval generalizes robustly to these "wild" prompts without modification.
* **Quantified Architectural Novelty:** Conducted a **Synonym Expansion Ablation** to address "incremental novelty" concerns. Results showed catastrophic performance drops (Spatial $\downarrow$ 75%, Numeracy $\downarrow$ 43%, Attribute $\downarrow$ 56%) without our module, proving that our open-vocabulary architecture is a critical necessity, not just an engineering improvement.

* **Clarified Efficiency and Scalability:** Provided a granular breakdown of latency vs. cost, explaining the trade-off between CPU-intensive attribute cropping and token-intensive numeracy reasoning. We confirmed that the full pipeline runs locally on consumer hardware with negligible marginal cost.
* **Refined Interpretability claims:** Clarified that our diagnostic feedback relies on deterministic, rule-based logic (Decompose $\rightarrow$ Detect $\rightarrow$ Verify) rather than opaque LLM hallucinations

We believe these revisions decisively address the main concerns regarding open-vocabulary claims, reproducibility, and synthetic bias, establishing SANEval as a robust and standardizable tool for Text-to-Image bechmarking.

We are happy to discuss and clarify any remaining questions.

Thanks again for your time,

**The SANEval Authors**

---

> ### Comment · Area_Chair_QxtC · 2025-11-28
>
> Dear reviewers,
>
> The authors have submitted their rebuttal.  Please carefully read the rebuttal and proactively participate in discussion with the authors.
>
> AC

---

### Meta-Review · Area_Chair_kcHL · 2026-01-07

**Summary:**

This paper introduces SANEval, a diagnostic benchmark for evaluating compositional faithfulness in text-to-image models across attributes, spatial relations, and numeracy. Reviewers agree the problem is important and the pipeline is well engineered, with modular design and structured feedback. The rebuttal strengthened the paper through an open-source implementation, a human-curated evaluation set, and ablations demonstrating the importance of key components. However, several core issues remain only partially addressed: the open-vocabulary claim lacks quantitative validation on long-tail categories, metric robustness to detection and parsing errors is not rigorously analyzed, and human evaluation still lacks fine-grained reliability metrics. Overall, the work is substantially improved but retains unresolved questions about metric correctness and long-tail reliability.

**Reviewer Concerns:**

The authors responded thoroughly and added substantial new experiments, including an open-source pipeline, a human-curated hard set, and meaningful ablations. Several major concerns regarding reproducibility, prompt realism, and computational cost are convincingly addressed. However, key issues remain insufficiently resolved, particularly the lack of quantitative validation for long-tail open-vocabulary claims and limited analysis of metric correctness and error propagation. Overall, the rebuttal is strong in effort and scope, but does not fully eliminate concerns about the reliability of the proposed evaluation metric.

**Reviewer Scores:**

The rebuttal would likely reduce concerns from the more mildly negative reviewers, as issues around reproducibility, proprietary dependence, prompt realism, and computational cost were addressed with new experiments and an open-source pipeline. However, the most critical reviewer would not change their overall negative assessment. The core objections—specifically the lack of quantitative validation for long-tail/open-vocabulary claims and the absence of rigorous analysis of metric correctness and error propagation—remain unresolved. As a result, while some negative views would soften, the strongest negative review would remain negative and would not convert to a positive recommendation.

---

### Decision · Program_Chairs · 2026-01-26

Reject